# On the link between stress field and small-scale hydraulic fracture growth in anisotropic rock derived from microseismicity

*Gischig, Valentin Samuel.[1]\*, Doetsch, Joseph [1], Maurer, Hansruedi [1], Krietsch, Hannes [1], Amann, Florian [1], Evans, Keith Frederick [1], Nejati, Morteza [1], Jalali, Mohammadreza [1], Valley, Benoît [2], Obermann, Anne Christine.[3], Wiemer, Stefan[3], Giardini, Domenico[1]*

[1]*Departement of Earth Sciences, ETH Zürich, Switzerland.*
[2]*Centre for Hydrogeology and Geothermics (CHYN), Université de Neuchâtel, Switzerland.*
[3]*Swiss Seismological Service, ETH Zürich, Switzerland.*
*\*Correspondance to: Valentin S. Gischig (gischig@erdw.ethz.ch)*

**Abstract** To characterize the stress field at the Grimsel Test Site (GTS) underground rock laboratory a series of hydrofracturing and overcoring tests were performed. Hydrofracturing was accompanied by seismic monitoring using a network of highly sensitive piezosensors and accelerometers that were able to record small seismic events associated with meter-sized fractures. Due to potential discrepancies between the hydrofracture orientation and stress field estimates from overcoring, it was essential to obtain high-precision hypocenter locations that reliably illuminate fracture growth. Absolute locations were improved using a transverse isotropic P-wave velocity model and by applying joint hypocenter determination that allowed computation of station corrections. We further exploited the high degree of waveform similarity of events by applying cluster analysis and relative relocation. Resulting clouds of absolute and relative located seismicity showed a consistent east-west strike and 70° dip for all hydrofractures. The fracture growth direction from microseismicity is consistent with the principal stress orientations from the overcoring stress tests, provided an anisotropic elastic model for the rock mass is used in the data inversions. $\sigma_1$ is significantly larger than the other two principal stresses, and has a reasonably well-defined orientation that is subparallel to the fracture plane. $\sigma_2$ and $\sigma_3$ are almost equal in magnitude, and thus lie on a circle defined by the standard errors of the solutions. The poles of the microseismicity planes also lie on this circle towards the north. Analysis of P-wave polarizations suggested double-couple focal mechanisms with both thrust and normal faulting mechanisms present, whereas strike-slip and thrust mechanisms would be expected from the overcoring-derived stress solution. The reasons for these discrepancies can be explained by pressure leak-off, but possibly may also involve stress field rotation around the propagating hydrofracture. Our study demonstrates that microseismicity monitoring along with high-resolution event locations provides valuable information for interpreting stress characterization measurements.

## 1 Introduction

Hydraulic fracturing (HF) is a method of creating artificial fracture networks in a rock mass by high-pressure fluid injection. It has become an essential technique in many underground engineering activities, including the enhancement of permeability in tight oil and gas reservoirs (Economides et al., 2000; Warpinski et al., 1998), and increasing the productivity of mines by fragmenting ore bodies (Jeffrey, 2000; Van As and Jeffrey, 2000). It is useful to distinguish between hydrofracturing (HF), and hydroshearing (HS). HS is a method of rock mass permeability enhancement that uses fluid injections to elevate pore pressure within the rock mass, thereby promoting the shear failure and attendant dilation and permeability increase of pre-existing fractures and faults that are close to critical stress. The amount of pore pressure increase required to initiate shear failure depends upon the degree of criticality (i.e., proximity to failure) of the discontinuity sets present in the reservoir, and is invariably less than required to drive new hydrofractures (Pine and Batcherlor, 1984; Kaiser et al., 2013). HS is often exploited in enhanced geothermal systems (e.g., Häring et al., 2008; Evans et al., 2005). Small-volume HF is also utilized in stress measurement (e.g., Haimson and Cornet, 2003; Hubbert and Willis, 1972), and is routinely used in many geological engineering projects where a detailed understanding of the stress state is needed to optimize the design of underground facilities (e.g., nuclear waste storage, gas storage, mining, tunneling, hydro-power facilities, etc; Zang and Stephansson, 2010). For stress characterization, boreholes are drilled into the rock mass and sections with no pre-existing fractures are isolated with hydraulic packers. After an initial pulse injection test to check the tightness of the packed-off interval, water is injected at a constant rate until the rock breaks down, i.e., a fracture initiates at the borehole wall. If the borehole is drilled sub-parallel to a principal stress direction, and deviations due to tensile strength anisotropy are not expected, then fluid pressure will tend to initiate an axial fracture at the boreholes wall in the direction of the maximum stress that acts normal to the borehole. Further complications can arise where the minimum principal stress is close to aligned with the borehole axis, and the preferred orientation of fracture propagation is in the plane normal to the borehole axis. In this case, the fracture can rotate from axial to lie normal to the minimum principal stress after propagating a short distance outside the wellbore stress concentration (Warren and Smith, 1985; Evans and Engelder, 1989), or even initiate as a transverse fracture (Evans et al., 1988). Subsequently, constant rate injections are repeated for several cycles to reopen and further propagate the fracture, commonly with periods of venting in between. Injection volumes in these small-scale hydrofracturing applications are usually on the order of $10 - 100$ liters (Haimson and Cornet, 2003). The pressure response is closely monitored to accurately record the pressure at which the breakdown occurs, and determine the instantaneous shut-in pressure (ISIP), both of which yield information on the local stress and rock stress conditions. The ISIP is the pressure prevailing once viscous pressure gradients have dissipated. For small volume treatments of importance here, it can be taken as the pressure required to just hold the fracture open, and is thus interpreted as a direct measure of the minimum principal stress magnitude $\sigma_3$.

High-pressure fluid injections, whether intended for hydrofracturing or hydroshearing, are invariably associated with microseismic events (or acoustic emissions). Such induced microseismicity can be used as a diagnostic tool to define the geometry and nature of failure of the individual events, regardless of HF scale (e.g., Ishida, 2001; Falls et al, 1992; Majer and Doe, 1986; Lockner and Byerlee, 1977). For this reason, microseismic monitoring is routinely used for monitoring stimulations of EGS reservoirs (Niitsuma et al., 1999), and more recently in oil and gas fracturing operations (e.g., Caffagni et al., 2016; Warpinski et al., 2012; Maxwell et al., 2010 ). At the other extreme of HF scale, it is also used to study the failure process of rock in laboratory tests (e.g., Chitrala, 2013)

During small-scale HFs, the orientation of the seismicity cloud is generally considered indicative of the fracture propagation directions, and thus is assumed to be normal to the minimum principal stress ($\sigma_3$) direction. Evidence comes from many small to intermediate-scale experiments in the laboratory and under in-situ conditions. Clouds of acoustic emissions in a salt mine observed by Manthei et al. (2003) indicate the local stress conditions and changes thereof. Majer and Doe (1986) showed in a laboratory field experiments that microseismicity clouds propagate perpendicular to the $\sigma_3$ direction. Recently, Chitrala et al. (2010) reported HF laboratory experiments on both isotropic sandstone and anisotropic pyrophyllite. They observed that fracture propagation is controlled by the stress orientation in isotropic rock, while in anisotropic rock the fracture orientation is also influenced by the anisotropy orientation. Similarly, laboratory investigations by Doe and Boyce (1989) showed that the stress orientation defines hydraulic fracture propagation only for a stress field anisotropy ratio $\sigma_1/\sigma_3 > 1.5$. At near isotropic stress conditions the fractures branch more strongly and without a preferred propagation direction, a phenomenom often referred as high fracture complexity (e.g., Katsaga et al., 2015). During *large-scale stimulations*, there is a tendency for seismic clouds to develop perpendicular to the minimum principal stress direction $\sigma_3$, (Häring et al., 2008; Evans et al., 2005) particularly for HF operations (e.g., Rutledge et al., 2004), although for HS stimulations in crystalline rocks there are many examples where the seismicity cloud is oblique to the $\sigma_3$ direction (e.g., Block et al., 2015; Murphy and Fehler, 1986; Pine and Batchelor, 1984), presumably reflecting the complex interplay between stress and the pre-existing fracture population that is suitably-oriented for slip reactivation. Furthermore, individual seismicity clusters within the overall seismicity cloud often strike oblique to the maximum principal stress (Eaton and Caffagni, 2015; Deichmann et al., 2014).

It is widely observed during large injections that most induced earthquakes show a double-couple mechanism, which can be taken to indicate that the seismic energy was produced by slip occurring along pre-existing fractures (Eaton and Mahani, 2015; Guilhem and Walter, 2015; Deichmann et al., 2014). Double-couple mechanisms are also observed during HF treatments (e.g., Chitrala et al., 2013; Ishida, 2001; Dahm et al., 1999), although the primary dislocation mechanism during HF is thought to be tensile fracturing (i.e., propagation in mode I or opening mode). Detailed moment-tensor analyses of the seismic waveforms have shown that most induced events involve a predominant double-couple

mechanism with relatively few indicating an occasionally strong tensile component (Horálek et al, 2010; Guilhem et al., 2014; Šílený et al, 2009; Martínez-Garzón et al., 2017). The wide-spread observation of dominant shear source characteristics of HF-induced microseismicity has been explained by fluid leak-off into small pre-existing fractures (Dusseault et al., 2011). Because tensile fracture opening is very inefficient in radiating seismic energy, the detected seismicity tends to be produced by slip along fractures adjacent to the growing hydrofracture. Thus, these shear events do not represent fracture growth themselves, but nonetheless serve to illuminate the overall plane of growth of the propagating HF.

Although the relationship between seismicity and HF growth is widely discussed in literature, there are few field-scale observations which investigate the relationship between meter-scale hydrofractures formed during stress tests and the ambient stress conditions (e.g., Zang et al., 2017; López-Comino et al 2017). In small-scale laboratory experiments the stress field is imposed to the samples and is precisely known (Chitrala et al., 2010; Doe and Boyce, 1989). In field cases, it is rare that two independent stress characterization methods are applied, even though this is desirable (e.g., Ask, 2009). For the hydrofracture method, the orientation of $\sigma_3$ is usually obtained from the orientation of the induced fracture, either from the azimuth of the trace at the wellbore obtained from imprint packers (Haimson and Cornet, 2003), or very rarely from the geometry of the microseismicity cloud (Zang et al., 2017; Majer and Doe, 1986). While simple fracture mechanical considerations suggest that hydrofractures should propagate in a plane normal to $\sigma_3$, in isotropic rock (e.g., Detournay, 2016), this is not necessarily the case for anisotropic rock, where theory and observations are sparse. To our knowledge, there are no published field-scale stress surveys which have combined independent methods to investigate the relationship between fracture growth derived from microseismicity and the stress field in an anisotropic rock mass.

In this study, we report on a microseismicity dataset recorded during three HF tests performed for stress field characterization in an underground research laboratory (i.e., the Grimsel Test Site). Independent stress measurements based on the overcoring method (Zang and Stephansson, 2010) were performed in the same borehole, and yielded comparable stress magnitudes but substantial differences in the orientation of $\sigma_3$. First, we describe the monitoring strategy and present the temporal evolution of seismicity in connection with the injection histories. Then, we apply anisotropic hypocenter localization including station corrections, as well as cluster analysis and relative localization. Further, we derive relative event magnitudes and focal mechanisms. The results are then compared to the overcoring stress field observations.

## 2 Experiment context and study site

### 2.1 The In-situ Stimulation and Circulation (ISC) experiment at the Grimsel Test Site (GTS)

The hydraulic fractures were created as part of a stress and rock mass characterization program that supports the design of a well-controlled and well-monitored hydraulic stimulation experiment, known as the In-situ Stimulation and Circulation (ISC) project (see Amann et al., 2017 for details). The core of this project is a series of injections of up to 1 m³ water into pre-existing faults to induce fault slip and fracturing. This is accompanied by an extensive monitoring program including measurements of strain, pressure and microseismicity. The ultimate goal of the experiment is to obtain novel insights into the fault stimulation processes that are essential for the technological development of enhanced or engineered geothermal systems (EGSs) and oil and gas well productivity enhancement. The experiments are performed at the Grimsel Test Site (GTS) in Switzerland (Figure 1) operated by the Swiss National Cooperative for the Disposal of Radioactive Waste (Nagra). The GTS is located at 1733 m above sea level and has an overburden of 400 – 500 m. The ISC experiment was performed between two tunnels (i.e., the VE and the AU tunnel), and the injection and monitoring boreholes were mostly drilled from the AU cavern at the southern end of the AU tunnel (Figure 1).

The host rock is the so-called Grimsel Granodiorite (GrGr), which changes into the Central Aar Granite (CaGr) about 50 m north of our experiment volume (Keusen et al., 1989). These rocks are part of the Aar Massive - a conglomerate of Variscan intrusions (age ~300 Ma) - that was later intruded by a network of lamprophyres and aplites around the study site. During the Alpine deformation phase, the magmatic rock body experienced greenschist-grade metamorphism and developed an Alpine foliation oriented roughly 140°/80° (dip direction / dip) on average. Apart from large-scale faults that are often overprinting metabasic dikes (i.e., metamorphose lamprophyres), the rock mass in the experiment volume is exceptionally intact, with only a few fracture sets present giving a net fracture density of 0 – 3 per meter.

*2.2 Stress field characterization*

Since in-situ stress is the relevant force driving fault slip induced during hydraulic stimulation, it is essential to define the local stress field. Thus, an extensive stress characterization program was performed that included both overcoring and hydraulic fracturing. Overcoring is a so-called stress relief method (e.g., Zang and Stephansson, 2010), during which a probe that measures radial strains and in some cases axial strains is inserted into a 38 mm diameter pilot hole. The hole is then overcored with a 96 mm (inner diameter) core bit thereby relaxing the stresses that prevailed within the rock surrounding the 38 mm diam. pilot hole. These stress-relaxation strains are measured by the probe and recorded. Two different probes were used in the stress characterization program. The first is the USBM probe (Zang and Stephansson, 2010) which measures diameter changes of the pilot hole in three directions, thereby defining the stress-relaxation strains in a plane normal to the borehole axis. Inversion of the strains using the measured elastic constants of the rock cylinder yields estimates of the three independent stress components (2D) in the plane normal to the borehole. The second is the CSIRO-HI probe (Worotnicki, 1993). When this probe is inserted into the pilot hole, glue is extruded

to bond an array of 12 axial and circumferential strain gages to the wall of the hole. Inversion of the 12 stress-relaxation strains using the appropriate elastic constants yields an estimate for the full 3D stress tensors (i.e., six components). A total of 16 overcoring experiments were carried out with 10 USBM and 6 CSIRO-HI probes.

The overcoring and hydrofracturing stress measurements were made in three boreholes. Two boreholes, SBH-1 and SBH-3, were drilled into the rock mass immediately to the south of the ISC experiment, where there are no faults and only a few fractures (0-3- fracture per borehole meters). The goal was to characterize the local stress conditions that are unperturbed by large-scale faults (i.e., several tens of meters away from any fault). The first borehole tested (SBH-1) was drilled sub-vertically (oriented 260°/75°) from the upper AU gallery (Figure 1, Table 1). It was intended to align with the best estimate of the sub-vertical principal stress towards the direction of the minimum principal stress component as estimated by Pahl et al., (1989) and Konietzky (1995), who found that axis minimum principal stress deviates from verticality. Four hydrofractures and three USBM overcoring tests were performed in SBH-1 with the goal of deriving the direction of the sub-horizontal stress components. The second borehole (SBH-3) was drilled sub-horizontally (190°/-5°, upwards inclined) towards the south from the AU cavern. Three hydrofracturing, three USBM, and three CSIRO-HI overcoring tests were conducted in this hole with the objective of measuring the sub-vertical stress component (hydrofracturing) as well as obtaining estimates of the full stress tensor (overcoring). A third sub-horizontal borehole (SBH-4, oriented 330°/-5°) was drilled towards NW-NNW from the AU cavern so as to penetrate one of the target fault zones of the ISC experiment. Four hydrofracturing, one HTPF (i.e., hydraulic testing of pre-existing fractures), three USBM, and three CSIRO-HI overcoring tests were performed in this hole with the aim of observing possible systematic stress field changes towards the fault zone. The hydrofracture (HF) tests in SBH-3 and SBH-4 were monitored with a microseismic monitoring system (due to technical issues microseismic monitoring during HF in SBH-1 was not possible). In this study, only microseismic events associated with the HF tests in SBH-3 are reported, as the monitoring layout proved to be ideal for recording high quality data. Results from SBH-4 will be reported in future work. A detailed presentation of all stress field investigations is provided by Krietsch et al., (2017). The main results are given in Table 1, and will be discussed in connection with microseismic observations in Section 5

210

*Table 1: Orientations of boreholes, fractures at the borehole wall, seismicity clouds, and principle stress orientations.*

| Dip direction and dip [°] | | |
|---|---|---|
| Borehole SBH-1 | 260 | 75 |
| Borehole SBH-3 | 190 | -5 |
| Foliation plane | 145 | 70 |
| *Fracture orientation from imprint packers* | | |
| SBH-1, 8 m | 158 | 82 |
| SBH-1, 11 m | 200 | 82 |
| SBH-1, 13m | 209 | 81 |
| SBH-1, 15 m | 173 | 79 |
| SBH-3, 18 m (HF1) | 143 | 71 |
| SBH-3, 13 m (HF2) | 139 | 71 |
| *Principal stress orientations* | | |
| $\sigma_{1, iso}$ | 141 | 06 |
| $\sigma_{2, iso}$ | 041 | 57 |
| $\sigma_{3, iso}$ | 235 | 33 |
| $\sigma_{1, aniso}$ | 093 | 37 |
| $\sigma_{2, aniso}$ | 190 | 10 |
| $\sigma_{3, aniso}$ | 293 | 51 |
| *Seismicity planes* | | |
| HF1 | 180 | 72 |
| HF2 | 175 | 76 |
| HF3 | 178 | 69 |
| Clusters | 172 | 69 |

## 3 Microseismic monitoring

### 3.1 Data acquisition

Monitoring microseismicity during meter-scale hydrofracturing requires high-sensitivity sensors. We used piezoelectric sensors similar to those commonly used in laboratory acoustic emission experiments (e.g., Ishida, 2002). They were designed by *Gesellschaft für Materialprüfung und Geophysik* (GMuG) for field deployment (Type GMuG Ma-Bls-7-70). These sensors are highly sensitive in the frequency range of 1 – 100 kHz, with the highest sensitivity at 70 kHz. They do not have a well-defined instrument response due to resonance peaks that depend upon sensor design and local installation to the rock (Kwiatek et al., 2011). Thus, ground velocity or acceleration cannot be derived readily. Because of this, the piezosensors at several locations were combined with calibrated one-component accelerometers (Type Wilcoxon 736T) that have a flat instrument response in the range ~2 Hz - 17 kHz.

The network layout is presented in Figure 1. A total of 28 piezosensors were used, 20 of which were clamped to polished rock faces at the tunnel wall. Five sensors were installed in each of the following locations: the VE tunnel (same level as AU cavern), in the staircase linking the AU cavern to the KWO tunnel, in the KWO tunnel, and in the upper AU gallery (16 m above AU cavern). The sensor

spacing is around 10 - 15 m. The sensors in the staircase, KWO tunnel, and upper AU gallery (sensors S6 – S20) were installed at blasted tunnel walls, which may have a more pronounced excavation damage zone than the ones (S1 – S5) at the mechanically excavated VE tunnel. At four of these sensor positions, accelerometers were glued to the rock next to the piezosensor. Additionally to the 20 sensors, a borehole sensor array with eight piezosensors and a sensor spacing of 1 m was deployed in borehole SBH-1 (diameter 101 mm). These sensors were pressed pneumatically against the borehole wall. The borehole sensors are the closest to the end of borehole SBH-3, and have a distance of ~9 m from the HF1 interval. The farthest away from the borehole are the sensors S1 - S5 with distance from 55 – 72 m. Note that only a few events were recorded at sensors with source – receiver distance larger than 30 m.

The sensors were digitized with a 32-channel acquisition system that records signals with 1 MHz sampling rate. Prior to digitization, the signals were high-pass filtered with corner frequencies of 1000 Hz and 50 Hz for the piezosensors and the accelerometers, respectively. The 32-channel system has a built-in event-detection and localization algorithm. At detection of an event, 32.768 ms (i.e., $2^{15}$ samples) of all traces including ~ 10 ms of pre-signal time are stored. Roughly, six event traces of ~32 ms can be stored per second implying that during some time after the events no further events can be detected and stored (i.e., because the system is occupied with storing the current waveform). This 'dead-time' of about 150 ms after each detected event implies that events occurring within this time cannot be detected and recorded. In case of continuous triggering, this would amount to a data loss of 80%. To be able to also detect events that may fall into this dead-time, and to recover small events not automatically detected, 16 selected channels were additionally recorded with a system that recorded data continuously without automatic event detection. Similar monitoring systems have been used in deep mines where they successfully recorded seismicity with magnitudes down to $M_w$-4.1 (Kwiatek et al., 2011; Plenkers et al., 2010), and in a recent HF experiment in an underground laboratory comparable to our experiment (Zang et al., 2017).

### 3.3 Joint hypocenter determination and cluster analysis

To obtain high-resolution event location from the microseismic data, the following workflow proved to be effective.

1. *Localization with isotropic velocity model and event filtering:* P-wave arrivals were manually picked. For this, traces were filtered with a band-pass filter with corner frequencies of 1 and 20 kHz. A first locating attempt assuming an isotropic homogeneous medium model with a P-wave velocity of 5150 m/s was performed to detect mis-picked first arrivals or events with unstable location solutions. Arrival times with large residuals and events with unstable location solutions were re-examined to ensure that no erroneous signals or phases were picked. Then, the following filtering criteria were applied: I) Arrival time with residuals >400 µs were removed. II) Events

with too few arrival time observations were removed. Note that although all events were best detected on the 8-sensor borehole sensor array, the array aperture is only 7 m and so an additional three arrival times at other piezosensors were required. If these were not available, the event was removed. III) Events for which localization did not converge after 200 iterations were removed. S-wave arrivals were not included in the location, because only few S-wave arrivals could be reliably picked, and the anisotropic S-wave velocity model is not well constrained.

2. *Deriving best-fit anisotropic velocity model:* With the remaining events, a transversely isotropic P-wave velocity model (i.e., based on the weak elastic anisotropy approximation of Thomson, 1986) was determined with a grid search algorithm that minimized the median residual RMS over all events. Thomson's formulation for transverse isotropy is:

$$v_P = v_{P,sym}(1 + \delta \sin^2(\theta)\cos^2(\theta) + \epsilon \sin^4(\theta)) \qquad (1)$$

Here, $v_{P,sym}$ is the P-wave velocity along the anisotropy symmetry axis (usually the minimum velocity) and $\theta$ the angle between the symmetry axis and the ray path. The Thomson parameter $\epsilon$ describes the relative increase of the velocity perpendicular to the symmetry axis, and $\delta$ defines the angular dependence of velocity (i.e., the 'shape' of velocity anisotropy). In our grid-search, we varied the symmetry-axis orientation, $v_{P,sym,}$ $\delta$ and $\epsilon$.

3. *Joint hypocenter determination (JHD)* (e.g., Maurer and Kradolfer, 1996). With this method, locations are not determined for each event individually. Instead, all events are jointly determined with a least-squares approach, in which also velocity model parameters and station corrections are computed. The latter denote systematic shifts in travel time arising from an error in sensor locations or geological conditions around the sensor (here for instance a pronounced excavation damage zone) that locally reduce the seismic velocity. The anisotropic JHD approach is described in detail in the Appendix. In our case, only station corrections were included in the inversion. The seismic velocity parameters were not computed as the clustered event distribution did not allow for a stable inversion, and because the velocity model was sufficiently constrained with the grid-search approach of Step 2.

4. *Location error estimation:* To compute the error of the source locations due to uncertainties in the manually picked arrival times, we perturbed the arrival times with a randomly distributed value with a standard deviation of 0.04 ms (i.e., 40 samples). The perturbed arrival times were used to compute new event locations. Repeating this 1000 times yields point clouds of a statistically representative number of possible event locations scattered around the locations determined from the unperturbed arrival times. Applying principle component analysis to these point clouds results in the three principle directions of the point cloud and the error along these (e.g., 95% quantiles or confidence intervals of the location components along the axes). In addition to the above event filter criteria, only events whose largest error axis was smaller than 2 m (i.e.±1 m) were used for analysis of the seismicity cloud geometry.

5. *Cluster analysis:* To better resolve details within the seismic clouds, cluster analysis and relative localization were performed following the approach described by Maurer and Deichmann (1995) or Deichmann et al. (2014). Cross-correlation between the P-waves was performed for all events and all stations to derive the correlation coefficient as a measure of waveform similarity and the corresponding lag time. The correlation coefficient of all stations of one event pair is combined as follows: first we apply the variance stabilizing Fisher transformation to the correlation coefficients, then average all transformed correlation coefficients above a threshold of 0.85, and finally apply the inverse Fisher-transform. Thus obtained averaged correlation coefficients can be combined in a correlation matrix showing the correlation between all event pairs. Event clusters were extracted using this matrix by assuming that similar events should exhibit similar row-patterns, i.e., events that strongly correlate should also correlate similarly with all other events. Events are assigned to a cluster if the correlation between the row-patterns are better than 0.98. These parameters were determined by trial-and-error.

6. *Arrival time adjustment:* For the events belonging to the extracted clusters the arrival times were adjusted using the approach suggested by Deichmann & Garcia-Fernandez (1992). At any station, the time-differences between events are optimized by considering the time-lags between each event pair of the cluster. To obtain an absolute time for each station and event, a master event has to be determined, to which all other arrival times are related to. We define the master event to be the one with the most P-wave arrivals. In case several events reached the maximum number of arrivals, the one with the largest median over all wave amplitudes was chosen.

7. *Relative relocation:* The adjusted travel times were used to relocate the events of each cluster using the absolute master event location as start value for the inversion.

**4 Results**

*4.1 Temporal evolution during hydrofracturing*

Our event catalogue consists of events from the 32-channel real-time event detection and of events extracted during post-processing from the continuous data recorded for 16 channels. All events were visually inspected to separate false triggers (e.g., electromagnetic high frequency or anthropogenic signals) from seismic signals induced by HF. The injection rate and pressure during the three hydro-fractures labeled HF1 (at 18 m borehole depth), HF2 (13 m depth), and HF3 (8 m depth) in borehole SBH-3 (see Figure 1) are shown in Figure 2 along with the cumulative number of detected events. In total 1161, 482 and 274 events were detected during HF1, HF2, and HF3, respectively. The difference in the number of detected events is most likely explained by the proximity of the sources to the borehole sensor array (9 m, 14 m, and 19 m respectively). These sensors were the most sensitive, possibly due to the lower noise-levels in the borehole (i.e., roughly less than half of the noise-level of

the tunnel sensors), and their surroundings bearing a greater resemblance to a full-space than applies to the tunnel-wall sensors. All detected events were at least recorded at the borehole sensor array.

Each HF experiment includes four injection cycles – a breakdown cycle (i.e., initiation of the fracture) followed by three fracture reopening cycles. In all three experiments, almost all seismic events occurred during fracture reopening, but only few events were associated with breakdown of the rock (similar as reported for HF experiments by Zang et al., 2017). Seismicity rates seem to depend on injection rate, even though injection rate is the same for the breakdown cycle and the first reopening cycles (i.e., 1 l/min) but seismicity rates are not (Figure 2). In contrast, seismicity rates do not depend on injection pressure, because they increase for each reopening cycle while pressure is comparable. Seismicity versus injected volume is explored in Figure 3. The injection volume was smallest for the breakdown cycle (0.5 liter for HF1 and 1 liter HF2 and HF3). Also, during the reopening cycles, very few events occurred during injection of the first 0.5 – 1 liter, after which seismicity rates strongly increased. Apparently, a minimum of 0.5 – 1 liters of injection volume is required to induce detectable seismicity, which was not reached during the break-down cycle although a fracture was clearly induced. Currently, the reason for such an aseismic fracturing phase below a threshold volume is not clear. Note that the relative event numbers after shut-in (i.e., grey lines) generally increases with every injection cycle; 5 – 10 % of all events occurred during the shut-in period of the second reopening cycle, 10 – 15% after the third reopening cycle.

### 4.2 Joint hypocenter determination (JHD)

After removing bad quality P-wave arrivals or events based on the aforementioned criteria (Steps 1 and 4 in Section 3.3), only 8% (88 events), 19% (92 events) and 25% (69 events) of all events of HF1, HF2, and HF3, respectively, met the criteria for JHD. The parameters used for JHD are given in Table 2. The anisotropic P-wave velocity model (Table 2) agrees well with estimates of seismic anisotropy from active seismic experiments at the GTS (see Doetsch et al., 2017; Vasco et al., 1998; Maurer and Green, 1997). The station corrections computed with JHD for both isotropic and anisotropic velocity models are shown for all sensor positions in Figure 4. In the isotropic case, the station corrections show systematic spatial patterns, as clearly seen for the borehole sensor array (Stations 21 – 28). These systematic distributions mostly disappear if anisotropy is considered. Also, the difference of the station corrections of the two velocity models shows that the impact of considering anisotropy is a change of the station corrections with a spatially systematic pattern. Thus, the station corrections strongly compensate for the angular velocity dependency, when an isotropic velocity model is used.

*Table 2: Anisotropic seismic velocity parameters used for JHD.*

| | |
|---|---|
| Seismic velocity $V_{P,sym}$ in direction of the symmetry axis | 5150 m/s |
| Thompson parameter $\varepsilon$ | 0.07 |
| Thompson parameter $\delta$ | 0.02 |
| Symmetry axis, azimuth | 330° |

| Symmetry axis, dip | 20° |

### 4.3 Seismicity distribution

The distributions of absolute event locations (derived with anisotropic JHD) are shown Figure 5. For HF1 and HF2, the seismicity clouds grow upwards from the injection locations (colored bars indicate packer intervals). The seismicity clouds show an oblate shape of almost 2 m width and lengths of the other two axes of 3.5 m and 5 m. The seismicity cloud of HF3 shows a downward migration and a slight offset to the injection locations (blue bar). Here, most seismicity is concentrated in a narrower band (< 1.5 m) than for HF1 and HF2. The diameter of the cloud is roughly 5 – 6 m. The orientation of the normal to the seismicity clouds are 0° ± 5° in azimuth for all three clouds, and 17°, 13° and 20° in dip for HF1, HF2, and HF3, respectively. There is a tendency for events that occurred during later injection cycles to be located further away from the injection point as the temporal pattern in Figure 5c-e shows. Similar observations were made by Baisch et al., (2009), who interpreted it as the 'Kaiser effect'. However, clear concentric rings of seismicity expected if seismicity only occurs around the propagating fracture tip are not observed. Possibly these rings are smeared to some degree due to limited location accuracy.

In Figure 6a, we show seismicity locations with the error bars, whereas Figure 6b shows the cumulative distribution functions of the errors along the three axes separately (i.e., the 95-percentiles along each axis). The latter includes events whose largest error exceeds 2 m, the cut-off limit of 2 m used for Figure 5 and 6a being indicated by the dashed line. About 25% of all located events have error limits > 2m. The median of the two-sided error along the three axes is 0.38 m, 0.72 m and 1.34 m. The orientation distribution of the largest error axis is shown in the stereographic projection (lower hemisphere) in Figure 6c, and indicates a predominant E-W azimuth (actually N100°E) of the largest error direction. Note that this closely corresponds to the direction of the largest extend of the seismicity clouds of HF1 and HF2 themselves, as can also be observed in Figure 6a.

The impact of considering anisotropy and station corrections on the shape and location of the seismicity clouds is illustrated in Figure 7a and b for the case of HF1. The largest differences are seen for locations derived using isotropic and anisotropic velocity models. Specifically, the seismicity cloud migrates towards east and upwards by 1 m on average if anisotropy is accounted for. In contrast, the impact of adding station correction is relatively minor; most events migrate by only a few decimeters. Generally, size and orientation of the seismicity clouds in Figures 7a and 7b do not change significantly in all comparisons; the lengths of the long axes of the clouds change by less than 0.5 m, and the orientations by less than 5°. We conclude that cloud size and orientation for all three HFs are robust results under the given location uncertainties. Nevertheless, considering anisotropy is important for the location of the seismicity cloud.

### 4.4 Cluster analysis and relative location

We found four clusters of events with highly similar waveforms as shown in Figure 8 for Station 9. In total 140 events out of a total of 249 locatable events were found to group in clusters. The largest cluster, denoted Cluster 1, includes 65 events. Note that each cluster does not necessarily consist of events from one hydrofracture, but may include events from all three hydrofractures, as is the case of Cluster 1. The waveforms in Figure 8 are aligned so that the corrected P-wave arrivals match. The high similarity of the P-waves among clustered events, but also between different clusters, is remarkable. We conclude that the fracturing mechanisms of all three fractures are partially similar, and - as expected from the essentially homogeneous rock mass – also the path effects on the wave are comparable. While the P-waves are very similar, the S-waves vary both in amplitude and arrival time. The differences in arrival times are explained by the differences in locations, i.e., an arrival time shift of 0.2 ms corresponds roughly to a hypocenter shift of 1 m. The variable S-wave amplitude compared to the P-wave amplitude possibly indicates that the sources may have a variable contribution of tensile components resulting in different S/P-wave amplitude ratios. In our case, observed S/P ratios (i.e., median over all sensors per event) range from 0.4 – 7.9. Based on theoretical considerations by (Eaton et al., (2014) who showed that tensile event have S/P ratios that do not exceed 4.617, we infer that events with large S/P ratios are shear-dominated, whereas those with low S/P ratios may have a significant tensile component. Similarly, Kwiatek and Ben-Zion, (2013) inferred the possible presence of tensile components using energy ratios of S- and P-waves. A more detailed analysis of S/P-wave amplitude ratios would require a better understanding of the spatial sensitivity to P- and S-waves of the piezosensors. This will be done in future work.

The events from each cluster were relocated relative to the master event highlighted in Figure 8. The resulting event distributions are shown in Figure 9. Compared to the absolute locations (i.e., from JHD), the clusters form much narrower discs (see also Figure 7c). The large axes of the discs cover nearly the entire area of the JHD-derived seismicity clouds. The orientation of the cluster discs only differs by about 5° in strike from the orientation of the–JHD-derived seismicity clouds. The cluster analysis did not reveal distinct sub-groups of events with geometric characteristics different to the overall seismicity cloud, such as found by Deichmann et al. (2014) and Phillips (2000). Instead, clusters contain events across all three fractures and the entire seismicity cloud, and thus helped confirming and refining the overall geometry of the fractures instead of resolving structures smaller than the fractures.

### 4.4 Relative magnitudes

We attempt to characterize the relative source strength by deriving a relative magnitude $M_r$ from the P-wave amplitudes. For that purpose, we adapt the concept used by Goebel et al. (2012) for laboratory

event magnitudes, but also account for seismic attenuation of the wave as was suggested by Zang et al (2017):

$$M_r = \log_{10}\left(\frac{1}{N}\sqrt{\sum_{i=1}^{N}(A_i \frac{r_i}{r_0} e^{\alpha(r_i - r_0)})^2}\right)$$

(2)

Here, $A_i$ is the maximum P-wave amplitude of the signal in time-domain filtered with a narrow band-pass filter between $3 - 7$ kHz, $r_i$ is the source-receiver distance, $r_0$ a reference distance (here chosen to be 10 m), N is the number of stations with a P-wave observation of the event. The parameter $\alpha = \pi f_0/(Q_P V_P)$ is the frequency-dependent attenuation coefficient, where $f_0$ is the dominant frequency, $V_P$ is the P-wave velocity, and $Q_P$ is quality factor representing aseismic attenuation. We corrected the amplitude $A_i$ following the strategy of Zang et al., (2017), using the middle frequency of the band-pass filter, which is $f_0 = 5$ kHz, and $Q_P = 30$ (Holliger and Bühnemann, 1996). The dominant frequencies (i.e., the maxima in the Fourier spectrum) in our case range from $1 - 10$ kHz.

Note that the magnitudes derived with this method have no absolute meaning and indicate source strength only relative to other events. To obtain a rough estimate of the recorded maximum magnitude, we compare theoretical spectra using the source model by Boatwright (1978) with the noise recorded at the accelerometers. Thus, we can roughly estimate an upper threshold of magnitudes of events observed at the accelerometers. Only three events were recorded by the accelerometer at sensor position S8, which is at a distance of each 12.3 m from the source with a poor signal-to-noise ratio of each ~3. In Figure 10a and b, the spectra of the three events are compared with noise spectra (converted to velocity from acceleration time series) typically recorded at S8. At around 2 kHz, the three events slightly emerge above the noise. Deriving absolute magnitudes from spectral fitting is not possible. Thus, we only attempt to derive a rough upper bound of the magnitudes by comparing theoretical source spectra to noise (Figure 10a and b). Considering stress drops 1 MPa (Figure 10a), we observed that the spectra of the three events fall in the band defined by the spectra corresponding to Mw-1.0 and -2.0, which would correspond to source radii of 4.3 m and 1.4 m, respectively. For a stress drop of 0.1 MPa (Figure 10b) the events fall in the band Mw-4.0 to -3.0. The corresponding source radii are within the range of range from 0.3 to 0.9 m. Thus, the magnitude of the events that were able to be recorded with the accelerometers (i.e., possibly the largest events in our sequence) is not well determined but possibly lies between Mw-3.5 and -1.5 depending on the assumed stress drop. However, the lower range of predicted source radii on the order of decimeters to meter are realistic considering that the hydrofractures span a few meters.

For all other locatable events, an *adjusted relative magnitude $M_{ra}$* by shifting all relative magnitudes obtained from equation 2 by the amount needed to give a value -2.5 for the largest event, thereby establishing approximate agreement with the mid-range estimate of magnitude $M_w$ of the event from

the accelerometer at S8. The resulting adjusted relative magnitudes are plotted in Figure 2g-i. Evidently, the $M_{ra}$ estimates tend to increase with increasing injection cycle. The sensitivity to weaker events is best for HF1, during which even $M_{ra}<-3.5$ could be located with an error better than 2 m (Figure 6b). Sensitivity degrades towards HF3, because the distance to the borehole sensor array increases. From Figure 10c, we observe that the three HFs were comparable in terms of magnitudes

distributions. The adjusted relative magnitudes $M_{ra}$ cover the narrow range from -3.5 to -2.5. The b-values of these sequences are overall quite high (b>2), in agreement with other HF studies (e.g., López-Comino et al., 2017), but are determined only for a narrow magnitude range and is thus uncertain.

*4.5 Focal mechanisms*

Only a few events showed clear P-wave onsets on sufficient sensors to yield the good directional coverage needed to obtain a usefully-constrained fault plane solution. Some examples are shown in Figure 11. Generally, two groups of events could be found: 1) events with compressive P-wave arrivals along the borehole array (located south of the HFs) and tensile arrivals at most of the sensors

above the HFs (in the upper AU gallery), and 2) events with the opposite pattern. In the first group, often a normal faulting or oblique normal faulting to oblique strike slip mechanisms could be fitted. A thrust faulting mechanism could be fitted for the events of the second group. In five out of nine cases, a focal plane could be fitted that perfectly or closely matched the cluster plane. For the thrust faulting events this was not possible.

It is noteworthy that the normal faulting style contradicts the stress field observations. As described in Section 5.2, the maximum principal stress $\sigma_1$ and $\sigma_3$ are sub-horizontal and $\sigma_2$ and $\sigma_3$ are very close in magnitude suggesting that a strike-slip to thrust fault mechanism is expected. Note that in many other induced seismicity studies most focal mechanisms were in agreement with the prevailing stress field, with only few events deviating from it (e.g., Baisch et al., 2015; Deichmann et al., 2014). Possibly, in

our case, a component of volumetric expansion or a compensated linear vector dipole (CLVD) mechanism modifies the pure double-couple mechanisms (Vavrycuk, 2011, Martínez-Garzón et al., 2017). Volumetric expansion would be consistent with growth of a tensile fracture driven by fluid injection. The double-couple mechanisms found here are in agreement with many studies that showed that seismicity associated with HF have double-couple sources (Chitrala et al., 2013; Dahm et al.,

1999; Nolan-Hoeksema & Ruff, 2001; Ohtsu, 1991).

**5 Discussion**

*5.1 Differences in HF and seismicity characteristics*

The three HF experiments are comparable regarding temporal evolution, seismicity cloud orientation and relative magnitude distributions. Nevertheless, HF3 differed somewhat from the other HFs in that it propagated downwards instead of upwards. HF3 also behaved differently in that the instantaneous shut-in pressure (ISIP) decreased with each cycle to stabilize 1 MPa lower than that of the others, and that the fluid volume recovery was markedly less (Figure 12). Indeed, there is a tendency of last-cycle

ISIPs, which are taken as direct measures of minimum principal stress, to decrease from ~9 MPa at the deepest measurement (HF1 at 18 m) to ~8 MPa for the shallowest (HF3 at 8 m). A similar decrease is also present in the breakdown pressures, which were 26.1 MPa for HF1, 25.7 for HF2 and for 23.4 MPa. While these slight differences may not be considered significant, the low volume recovery rate of HF3 is noteworthy. Less than 5 - 15 % of the total injected volume was recovered as opposed to

HF1 and HF2, for which it was 60 – 75 %. Low volume recovery rates indicate that a pre-existing permeable fracture network may have been intersected by the propagating HF. The Optical Televiewer (OPTV) images of the hydrofracture intervals shown in Figure 12c indicate that all three were free of pre-existing fractures. However, in case of HF3, a two-centimeter-wide dark band of biotite-rich zones can be observed about 10 – 20 cm further downhole. Upon revisiting the exact same interval 1.5 years

later (6 February 2017), it was not possible to reopen any fracture. Only when the packer interval was moved 0.3 m downhole could fluid be injected in the manner expected for fracture reopening, with pressures comparable to the initial test. It is also noteworthy that no fracture was detected in the imprint packer survey of the interval that was conducted after hydrofracturing. Although the biotite-bands are oriented parallel to foliation (150°/75°) and not parallel to the seismicity cloud (180°/70°),

they may have served as weakness zones that were reactivated during the hydrofracture test because water was able to penetrate sufficiently far along the packer seat. This explanation is also consistent with the fact that the seismicity cloud was offset towards south (i.e., downhole) by a few decimeters (Figure 5a and 9). The low recovery rate could be explained either by the packer sealing of the fracture again after releasing pressure from the interval, or by flow to the far field within the permeable

structure accessed by the biotite bands.

*5.2 Comparison to overcoring stress measurements*

Alongside HF, overcoring surveys were performed in all three boreholes as an independent stress characterization method (see Section 2.2). Out of six CSIRO-HI overcoring experiments, three were judged to have provided high-quality internally-consistent results (Bouffier et al., 2015). One of these

obtained at a depth of 9 m in SBH-3 was rated good (i.e., confidence level 4 on a total scale of 5), and the other two at depths of 9.2 m and 14.9 m in SBH-4 were ranked 5/5 and 4/5 respectively. Strain data from these three tests were inverted using two elastic models: an isotropic model and transversely

isotropic model (Krietsch et al., 2017). The elastic parameters for the models were constrained using numerical modeling to reproduce the strains recorded during bi-axial tests conducted on the instrumented cores immediately after extraction, and supplemented by laboratory tests. The stress tensors obtained from the inversions are presented in Figure 13 (values given in Table 1). If an isotropic elasticity model is used, there is a clear discrepancy between stress field orientations from overcoring and the planes of HF-induced seismicity: for it is expected that $\sigma_3$ would be normal to and $\sigma_1$ and $\sigma_2$ to be parallel to the seismicity plane. However, $\sigma_1$ is sub-horizontal and subtends an angle of about 60° with respect to the seismicity plane (poles included in Figure 13). Also, neither $\sigma_3$ nor $\sigma_2$ is normal to the seismicity plane. For the transversely isotropic model, Krietsch et al (2017) performed inversions for a range of parameter sets and showed that the degree of anisotropy (i.e., the ratio of the Young's moduli normal to and in the plane of the foliation) had the greatest influence on the principal stress orientations. Using a ratio of two suggested by laboratory tests, the orientation of $\sigma_1$ became 90°/35° (dip direction /dip), which is sub-parallel to the seismicity planes. The magnitudes of $\sigma_2$ and $\sigma_3$ are very close, with a difference of less than 2 MPa. As a consequence, small variations in the assumed elastic parameters produce strong variations of the orientation estimates for $\sigma_2$ and $\sigma_3$, the solutions for both extending almost completely around the circle normal to $\sigma_1$ (half-circle in Figure 13). Thus, uncertainties in the parameters defining the transverse anisotropic model preclude a unique determination of the $\sigma_3$ direction. However, the three hydrofractures showed consistent orientations lying along the circle defined by the solutions for the $\sigma_2$ and $\sigma_3$ orientations. We conclude that $\sigma_3$ is sub-horizontal oriented north-south and is sufficiently smaller than $\sigma_2$ that it defines consistent fracture growth directions. Thus, we have shown that microseismic monitoring in this case provides essential information for obtaining a conclusive stress tensor estimate.

Also included in Figure 13 are the orientations of the HF initiated at the borehole wall, as determined from oriented imprint (or impression) packer surveys (IP). Successful imprints of the traces of the induced fractures were obtained only for HF1 and HF2 in SBH-3. The absence of a trace for HF3 may be because the fracture initiated some decimeters downhole of the interval, as mentioned earlier (see Section 5.1). The traces of both fractures have orientations that are close to the foliation plane, which has a markedly different orientation to that of the seismicity clouds. The poles of the fracture traces obtained from imprint packer surveys of the four HF intervals in the sub-vertical SBH-1 borehole are also shown in Figure 13. These fractures scatter within a ±20° range, and match the seismicity cloud orientations on average.

In SBH-3, the foliation and its relative orientation with respect to the borehole may play a role both in influencing near-wellbore stress concentrations and in fracture initiation along the weak direction. The initiation of hydrofractures is controlled by the stress state around the wellbore, which includes a contribution from the steadily-increasing wellbore fluid pressure, and by defects and cracks in the borehole wall. Once a fracture is initiated, it enters a regime in which its growth is dominated by

fracture toughness and thus may deviate from local principal stress orientations. After this initial stage, the fracture gradually reorients to become aligned with the direction preferred by the principal stress directions. The reorientation process of hydro-fractures is controlled by many factors including fluid properties, injection rate, or stress field anisotropy (Zhang et al. 2011). Experimental evidence shows that anisotropic behaviour in crystalline rock is often the result of micro-cracks that have a preferred orientation parallel to the foliation plane (Nasseri and Mohanty, 2008). Such a micro-structure can produce anisotropy ratios of elasticity, strength and fracture toughness as large as two (Nasseri et al. 2010; Dai et al. 2013). Possibly, in our case, these micro-cracks have served as defects or weakness zones at which fractures could initiate. It seems that fractures initiated from flaws within the foliation plane, and propagated initially within this plane both radially and around the borehole. Beyond the toughness-dominated regime, fracture reorientation with respect to the principle stress directions occurred. Since this reorientation was not observed in the seismic clouds, it would seem to have occurred within a few decimetres. Assuming the stress magnitudes to be $\sigma_1 = 13 - 17$ MPa, $\sigma_2 = 8.5 - 9.5$ MPa and $\sigma_3 = 8.5$ MPa as proposed by Krietsch et al, 2017, the normal stress on the foliation plane in the far-field of the borehole is $\sigma_n = 8.9 - 9.4$ MPa. Thus, despite the small difference between the normal stress on the foliation plane and $\sigma_3$, it was easier for the fracture to cut through foliation instead of propagating along the foliation plane.

Another noteworthy feature of our seismicity clouds is the asymmetric growth of the HFs around the injection interval. Dahm et al. (2010) suggested that asymmetric bidirectional fracture growth during injection and bidirectional to unidirectional growth after shut-in may be driven by gradients of in-situ stress or pore pressure. In our case, fractures grow upwards in case of HF1 and HF2 and downwards in case of HF3, which would imply that a presumed stress or pressure gradient would change direction between 13 and 8 m borehole depths. We also observe unidirectional rather than bidirectional growth. Thus, we argue that the mechanism proposed by Dahm et al. (2010) might not be sufficient to explain asymmetric fracture growth in our case. To date, it is not entirely clear why fractures grew in such unidirectional manner.

It has been observed in various studies that fracture propagation in foliated rock can lead to a mixture of tensile failure mechanisms and shear mechanisms (e.g., stepped failure geometry) (Debecker and Vervoort, 2009). From our observations we cannot infer or exclude the existence of the tensile failure mechanism. However, the focal mechanism solutions observed for only a few events are a mixture of normal faulting (with some focal planes nearly parallel to the seismicity cloud) and thrust faulting. From our stress field estimates, we would expect strike slip (and possibly thrust faulting) mechanisms reflecting slip along optimally-oriented pre-existing fractures that intersect the HF plane. We argue that focal mechanisms are expected to be in agreement with the stress field orientation if the slip direction is governed only by the locally-uniform ambient stress field. Hence the variability of the mechanisms we observe must be due to additional factors. Nolen-Hoeksema & Ruff, (2001) proposed

three mechanisms that may produce seismicity during hydrofracturing. 1) tensile fracturing at the tip of the propagating fracture, 2) pressure leak-off into pre-existing fractures that intersect the propagating hydrofracture, resulting in their weakening and shear failure (e.g., Rutledge et al., 2004), or 3) slip along pre-existing fractures near the fracture tip induced by stress perturbations associated with fracture propagation (Martínez-Garzón et al., 2013, Warpinski and Branagan, 1989). In our case, pure tensile fracturing (mechanism 1) can be excluded for the observed double-couple events. To explore the other two mechanisms, the shear and normal stress acting on the focal planes in Figure 11 (i.e., the red half-circles) were computed using the stress field estimate by Krietsch et al., (2017), and the values plotted in the Mohr-Coulomb diagram shown in Figure 14. It is evident that overpressures of 7 – 9 MPa are able to explain slip along all observed focal planes. Thus, pressure-induced slip resulting from fluid leak-off (mechanism 2) can lead to diverse focal mechanisms as it permits structures that are not optimally-oriented in the stress field to be reactivated. It is also possible that stress field perturbations arising from the propagating hydrofractures (mechanism 3) may additionally promote criticality along these planes, although this is not resolved by the present observations. In this regard, the assumption of stress homogeneity within the study volume inherent in the shear and normal stress estimates plotted in Figure 14 may well be too simplistic. The presence of stress heterogeneity, either pre-existing or generated during the injections through mechanism 3 could potentially modify these values.

**6 Conclusion**

A series of HF tests were performed as part of a stress characterization survey at the Grimsel Test Site. The installation of a microseismic monitoring system proved valuable for studying the HF process on scales of decimeters to meters. The implemented workflow illustrates that many standard seismological tools – such as joint hypocentre location with station corrections, high-precision relative relocations of event clusters with similar waveforms, and focal mechanism analysis – can be applied to seismicity at such scales. For other seismological observables such as magnitudes, more efforts are required to obtain meaningful results and assess their uncertainties. In the present case, micro-seismic monitoring during the hydrofracture experiments proved crucial for the combined interpretation of the results of the stress characterization methods. The three hydrofracture operations in the SBH-3 borehole produced three flattened seismic structures that extended from at or close to the injection intervals. The structures had an EW strike and dipped at about 70° to the south. The overcoring strains inverted using an isotropic elastic model yielded stress tensor solutions whose minimum principal stress, $\sigma_3$, deviated significantly from normal to the seismic structures, as would be expected if the hydrofractures grow normal to $\sigma_3$. The discrepancy could be resolved by using a transversely isotropic elasticity model whose parameters were consistent with laboratory measurements on the core. Imprint packer surveys of the injection intervals in SBH-3 showed that the fractures initiated at the borehole

wall within the foliation plane, whose orientation differs significantly from that of the seismic structures. We interpret this to indicate that fracture nucleation occurred on flaws that lay in the foliation plane, and that the fracture initially extended within this weakness plane before rotating to lie normal to the minimum principal stress after propagating at most several decimeters. Focal

mechanisms show a mixture of normal faulting and thrust faulting mechanisms, whereas a strike-slip mechanism, or possibly thrust, is expected from the stress field orientation. It is conceivable that stress perturbation and pressure leak-off around the propagating fracture strongly influences the source mechanism of the seismic events. Our observations illustrate the challenges faced in stress characterization surveys in moderately anisotropic rock; a combination of overcoring, HF, and micro-

seismic monitoring were essential to arrive at a conclusive interpretation of the all observations.

**Acknowledgments**

The ISC is a project of the Deep Underground Laboratory at ETH Zurich, established by the Swiss Competence Center for Energy Research - Supply of Electricity (SCCER-SoE) with the support of the

660 Swiss Commission for Technology and Innovation (CTI). Funding for the ISC project was provided by the ETH Foundation with grants from Shell and EWZ and by the Swiss Federal Office of Energy through a P&D grant. Hannes Krietsch is supported by SNF grant 200021_169178. The Grimsel Test Site is operated by Nagra, the National Cooperative for the Disposal of Radioactive Waste. We are indebted to Nagra for hosting the ISC experiment in their GTS facility and to the Nagra technical staff

for onsite support. We also thank G. Kwiatek and E. Caffagni and an anonymous reviewer for their valuable comments and corrections that helped improving the manuscript. All data are available through the link doi:10.3929/ethz-b-000217536.

**Appendix: Earthquake location using an anisotropic velocity model**

In the following, we derive the analytical derivatives used for the Jacobi matrix for earthquake location considering transverse isotropic P-wave velocity. In the joint hypocenter determination, the inverse problem to be solved involves minimizing the discrepancy between the observed and predicted

arrival times, that is, $\min\left\{\left\|\mathbf{t^{obs}} - \mathbf{t^{calc}}\right\|^2\right\}$ by finding an appropriate set of model parameters

(1)    $\mathbf{m} = \left(s_j^x, s_j^y, s_j^z, t_j^0, t_i^S\right)$.

Here, the $s_j^x$, $s_j^y$ and $s_j^z$ are the hypocentral coordinates of the $j$th event, $t_j^0$ the source time, and the $t_i^S$ station correction at the sensor position $i$. $t_{ij}^{obs}$ denote the arrival time picks as where the index $i$ runs from $1$ to the total number of picks $N_j$, of the event $j$. $j$ runs from 1 to the total number of events $M$. They can be collected in a vector $\mathbf{t^{obs}}$. The predicted travel times $t_{ij}^{calc}$ can be computed as:

(2) $\qquad t_{ij}^{calc} = \dfrac{l_{ij}}{v} + t_j^0 + t_i^S$

$l_{ij}$ is the length of the entire ray path between the $i$th sensor and the hypocenter of the $j$th event. The inverse problem requires the derivatives $\dfrac{\partial t_{ij}^{calc}}{\partial s_j^x}$, $\dfrac{\partial t_{ij}^{calc}}{\partial s_j^y}$, $\dfrac{\partial t_{ij}^{calc}}{\partial s_j^z}$, $\dfrac{\partial t_{ij}^{calc}}{\partial t_j^0}$ and $\dfrac{\partial t_{ij}^{calc}}{\partial t_i^S}$ to be computed.

The partial derivative with respect to the origin time $t^0$ is

(3) $\qquad \dfrac{\partial t_i^{calc}}{\partial t^0} = 1$.

Similarly, the partial derivative with respect to the station correction $t_i^S$ is

(4) $\qquad \dfrac{\partial t_i^{calc}}{\partial t_i^S} = 1$

The derivatives with respect to ($s_j^{x,y,z}$) are computed by considering equation (2). Let us assume that each ray segment $l_{ij}$ is bound the $j$th hypocenter ($s_j^x, s_j^y, s_j^z$), and the $i$th receiver location ($r_{ij}^x, r_{ij}^y, r_{ij}^z$). The length of a segment is equal to (ignoring the index $j$ in the following):

(4) $\qquad l_i = \sqrt{\left(r_i^x - s^x\right)^2 + \left(r_i^y - s^y\right)^2 + \left(r_i^z - s^z\right)^2}$ .

Only the first term of the sum in equation (2) contributes to the derivatives with respect to the hypocentral coordinates (only in the first term the hypocentral parameters ($s^x, s^y, s^z$) are involved). Unlike in the isotropic case, however, not only the segment $l_i$ contributes to the derivatives with respect to ($s^x, s^y, s^z$), but also the velocity $v = v(s^x, s^y, s^z)$, which become dependent of the take-off angle of the incoming ray. Therefore, we can write:

(5) $\qquad \dfrac{\partial t_i^{calc}}{\partial s^{x,y,z}} = \dfrac{1}{v\left(s^{x,y,z}\right)} \dfrac{\partial l_i}{\partial s^{x,y,z}} - l_i \dfrac{\partial v}{v^2 \partial s^{x,y,z}}$ .

Here, the derivative in the first term is (considering first $s^x$):

$$(6) \quad \frac{\partial l_i}{\partial s^x} = \frac{\partial \sqrt{\left(r_i^x - s^x\right)^2 + \left(r_i^y - s^y\right)^2 + \left(r_i^z - s^z\right)^2}}{\partial s^x}$$

$$= \frac{-2\left(r_i^x - s^x\right)}{2\sqrt{\left(r_i^x - s^x\right)^2 + \left(r_i^y - s^y\right)^2 + \left(r_i^z - s^z\right)^2}},$$

$$= \frac{-\left(r_i^x - s^x\right)}{l_i}$$

Similarly,

$$(7) \quad \frac{\partial l_i}{\partial s^y} = \frac{-\left(r_i^y - s^y\right)}{l_i}, \text{ and}$$

$$(8) \quad \frac{\partial l_i}{\partial s^z} = \frac{-\left(r_i^z - s^z\right)}{l_i}.$$

The expressions for the spatial derivatives in equations (6) to (8) can also be expressed with angles $\alpha_i$ and $\beta_i$ that denote the azimuth and the inclination of the ray path leaving the hypocenter. The resulting solid angle represents the so-called *take-off angle*.

Equations (6) to (8) can be rewritten in terms of the angles $\alpha_i$ and $\beta_i$:

$$(9) \quad \frac{\partial t_i^{calc}}{\partial s^x} = -\cos(\alpha_i)\cos(\beta_i).$$

$$(10) \quad \frac{\partial t_i^{calc}}{\partial s^y} = -\sin(\alpha_i)\cos(\beta_i).$$

$$(11) \quad \frac{\partial t_i^{calc}}{\partial s^z} = -\sin(\beta_i).$$

For the derivative in the second term of equation (5), we have to assume an anisotropic P-wave velocity model. We here use the Thomsen parameterization for weak anisotropy (Thomson, 1986):

$$(12) \quad v = v_{P,sym}\left(1 + \delta \sin^2(\theta)\cos^2(\theta) + \varepsilon \sin^4(\theta)\right),$$

where $\theta$ is defined as the angle between the symmetry axis of the anisotropic medium, oriented along $\phi^{sym}$ and the ray segment $l_i$, oriented along $\phi^{ray}$, that is,

$$(13) \quad \theta = \arccos\left(\phi^{sym} \cdot \phi^{ray}\right),$$

with

(14) $\phi^{sym} = \begin{pmatrix} \cos\phi_{inc}^{sym} \cos\phi_{azi}^{sym} \\ \cos\phi_{inc}^{sym} \sin\phi_{azi}^{sym} \\ \sin\phi_{inc}^{sym} \end{pmatrix}$

(*inc* = inclination angle, *azi* = azimuth), and

(15) $\phi^{ray} = \begin{pmatrix} \cos\beta_i \cos\alpha_i \\ \cos\beta_i \sin\alpha_i \\ \sin\beta_i \end{pmatrix}.$

For determining the derivatives $\dfrac{\partial v}{\partial s^{x,y,z}}$, we define

(16) $\Psi = \left(1 + \delta\sin^2(\theta)\cos^2(\theta) + \varepsilon\sin^4(\theta)\right),$

and use the chain rule

(17) $\dfrac{\partial v}{\partial s^{x,y,z}} = \dfrac{\partial v}{\partial \Psi}\dfrac{\partial \Psi}{\partial \theta}\dfrac{\partial \theta}{\partial s^{x,y,z}}$

with

(18) $\dfrac{\partial v}{\partial \Psi} = v_{P,sym}$

(19) $\dfrac{\partial \Psi}{\partial \theta} = 2\delta\left(\sin(\theta)\cos^3(\theta) - \cos(\theta)\sin^3(\theta)\right) + 4\varepsilon\sin^3(\theta)\cos(\theta)$

(20) $\dfrac{\partial \theta}{\partial s^x} = \dfrac{\partial}{\partial s^x}\cos^{-1}\left(\begin{pmatrix} \cos\phi_{inc}^{sym} \cos\phi_{azi}^{sym} \\ \cos\phi_{inc}^{sym} \sin\phi_{azi}^{sym} \\ \sin\phi_{inc}^{sym} \end{pmatrix} \cdot \begin{pmatrix} \cos\beta_i \cos\alpha_i \\ \cos\beta_i \sin\alpha_i \\ \sin\beta_i \end{pmatrix}\right)$

Setting

(21) $\Gamma = \begin{pmatrix} \cos\phi_{inc}^{sym} \cos\phi_{azi}^{sym} \\ \cos\phi_{inc}^{sym} \sin\phi_{azi}^{sym} \\ \sin\phi_{inc}^{sym} \end{pmatrix} \cdot \begin{pmatrix} \cos\beta_i \cos\alpha_i \\ \cos\beta_i \sin\alpha_i \\ \sin\beta_i \end{pmatrix}$

$= \cos\phi_{inc}^{sym} \cos\phi_{azi}^{sym} \cos\beta_i \cos\alpha_i + \cos\phi_{inc}^{sym} \sin\phi_{azi}^{sym} \cos\beta_i \sin\alpha_i + \sin\phi_{inc}^{sym} \sin\beta$

we can write (considering equation 6 – 11)

$$\frac{\partial \theta}{\partial s^x} = -\frac{1}{\sqrt{1-\Gamma^2}} \frac{\partial \Gamma}{\partial s^x}$$

$$= -\frac{1}{\sqrt{1-\Gamma^2}} \frac{\partial}{\partial s^x} \left( \cos\phi_{inc}^{sym} \cos\phi_{azi}^{sym} \frac{r_i^x - s^x}{l_i} + \cos\phi_{inc}^{sym} \sin\phi_{azi}^{sym} \frac{r_i^y - s^y}{l_i} + \sin\phi_{inc}^{sym} \frac{r_i^z - s^z}{l_i} \right)$$

$$(22) \qquad = \frac{1}{\sqrt{1-\Gamma^2}} \left( \begin{array}{l} \dfrac{\cos\phi_{inc}^{sym}\cos\phi_{azi}^{sym}}{l_i} + \\[2mm] \left\{ \cos\phi_{inc}^{sym}\cos\phi_{azi}^{sym}\dfrac{r_i^x - s^x}{l_i^2} + \cos\phi_{inc}^{sym}\sin\phi_{azi}^{sym}\dfrac{r_i^y - s^y}{l_i^2} + \sin\phi_{inc}^{sym}\dfrac{r_i^z - s^z}{l_i^2} \right\}\dfrac{\partial l_i}{\partial s^x} \end{array} \right).$$

$$= \frac{1}{\sqrt{1-\Gamma^2}} \left( \frac{\cos\phi_{inc}^{sym}\cos\phi_{azi}^{sym}}{l_i} + \frac{\Gamma}{l_i}\frac{\partial l_i}{\partial s^x} \right)$$

$$= \frac{1}{l_i\sqrt{1-\Gamma^2}} \left( \cos\phi_{inc}^{sym}\cos\phi_{azi}^{sym} - \Gamma\cos\alpha_i\cos\beta_i \right)$$

Similarly, we find

$$(23) \qquad \frac{\partial \theta}{\partial s^y} = \frac{1}{l_i\sqrt{1-\Gamma^2}} \left( \cos\phi_{inc}^{sym}\sin\phi_{azi}^{sym} - \Gamma\sin\alpha_i\cos\beta_i \right),$$

and

$$(24) \qquad \frac{\partial \theta}{\partial s^z} = \frac{1}{l_i\sqrt{1-\Gamma^2}} \left( \sin\phi_{inc}^{sym} - \Gamma\sin\beta_i \right)$$

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

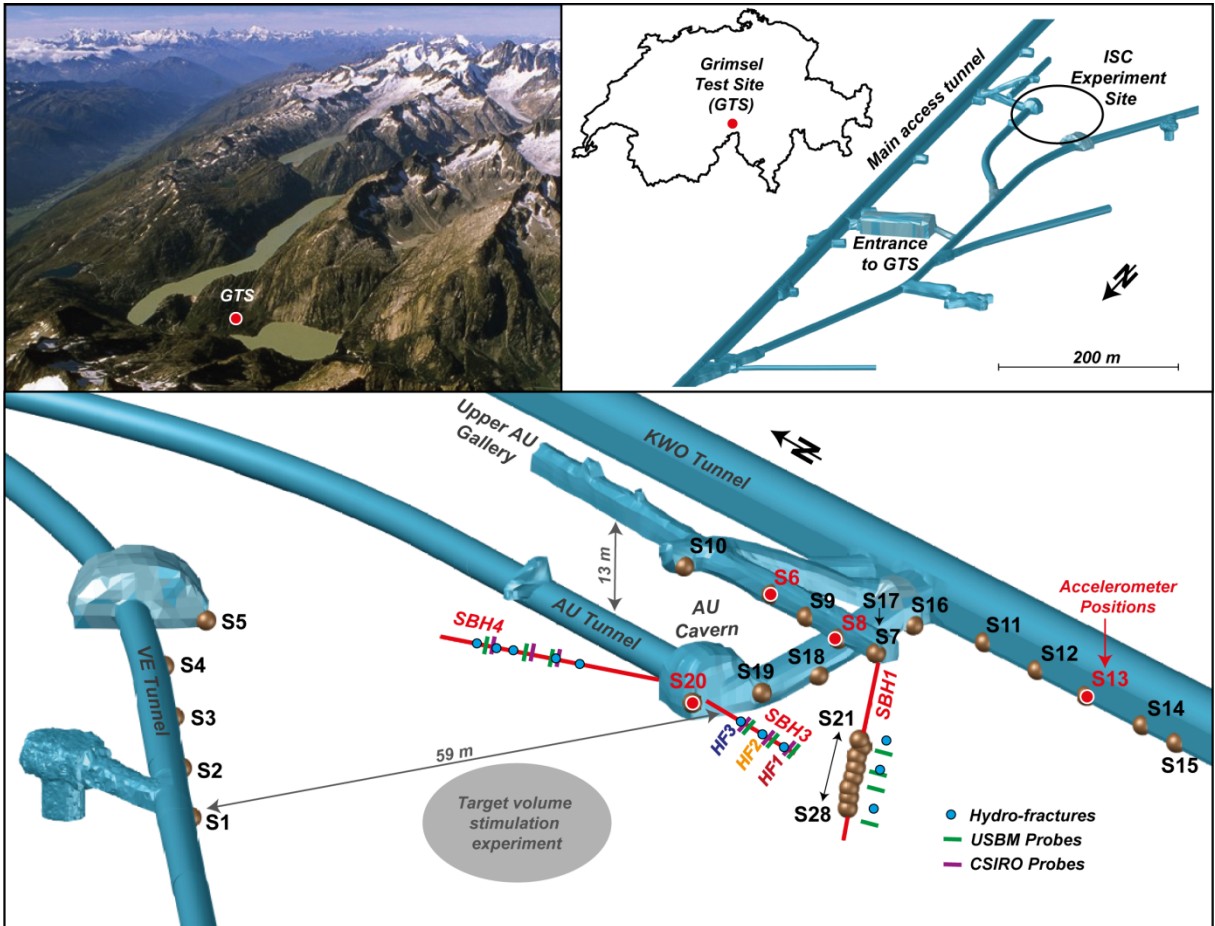

*Figure 1: The study site is located in the Bernese Alps in southern Switzerland (photo from www.grimselstrom.ch/elektrische-energie/kraftwerke-und-stauseen), and consists of a network of tunnels, with the ISC experiment site located between two tunnels. The stress characterization survey used three boreholes (SBH-1, 3 and 4) in which overcoring (using USBM and CSIRO probes) and hydraulic fracturing were performed. S1-S28 mark the seismic sensors.*


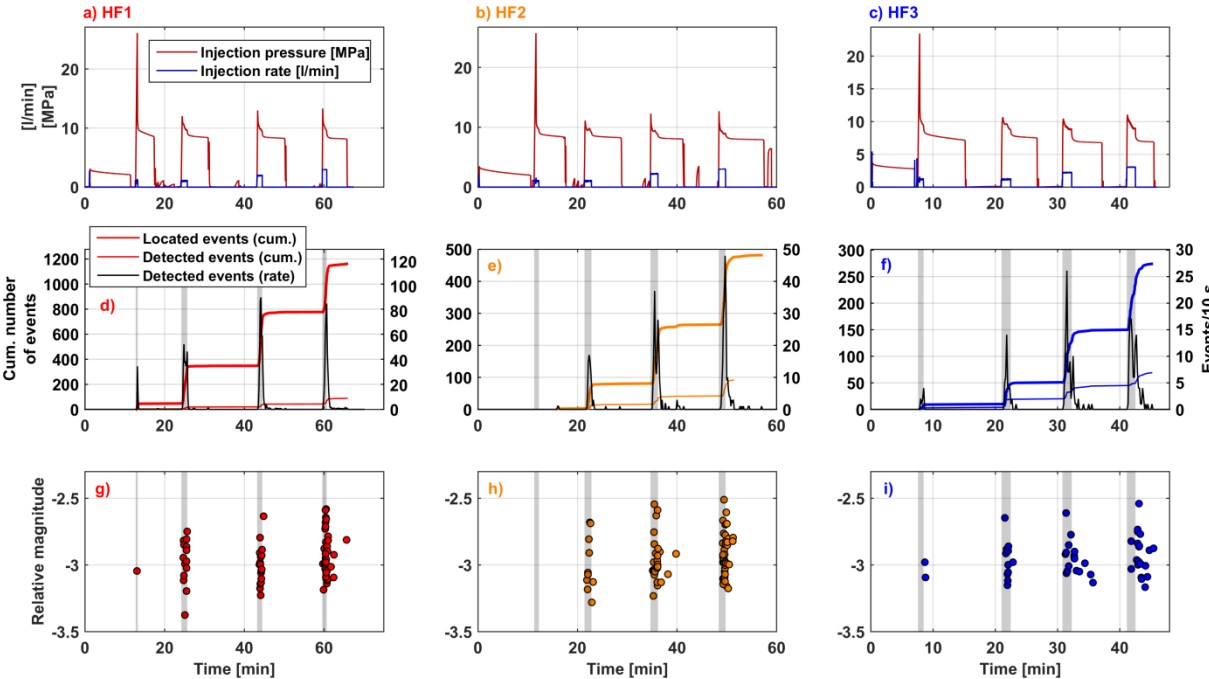


***Figure 2:*** *Temporal evolution of seismic events during hydrofracturing tests in SBH-3. Panels a-c show injection rate and pressure, d-f show the cumulative number of events, and g-i show the adjusted relative magnitude. Events occur mostly during injections (gray shaded areas), but some events occur after shut-in.*



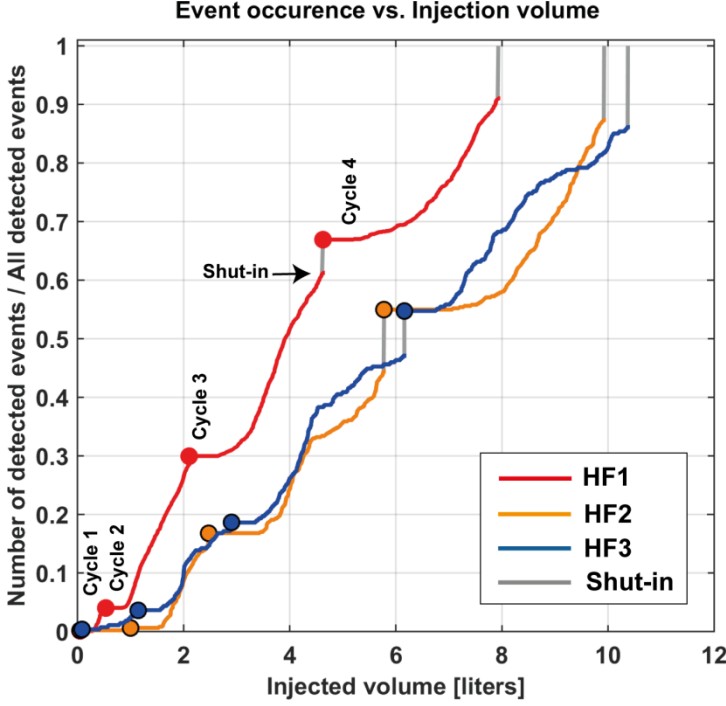

*Figure 3:* *Cumulative fraction of events as a function of cumulative injected volume for hydro-factures*
*HF1-3.*



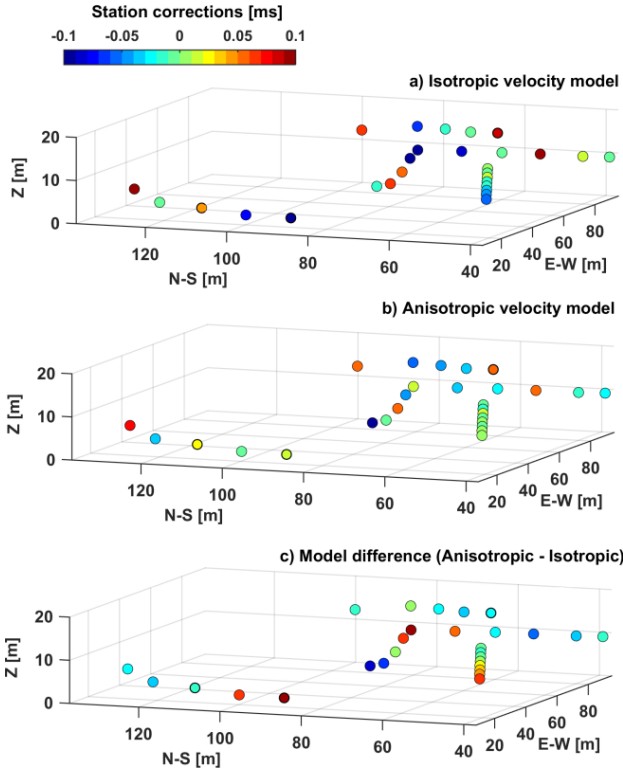

*Figure 4:* *Sensor distribution and corresponding station corrections. a) Station correction for an isotropic velocity model. b) Station corrections for an anisotropic velocity model. c) Difference between station corrections of the two velocity models. It shows the part of the station corrections using an isotropic velocity model that compensates for neglecting anisotropy.*

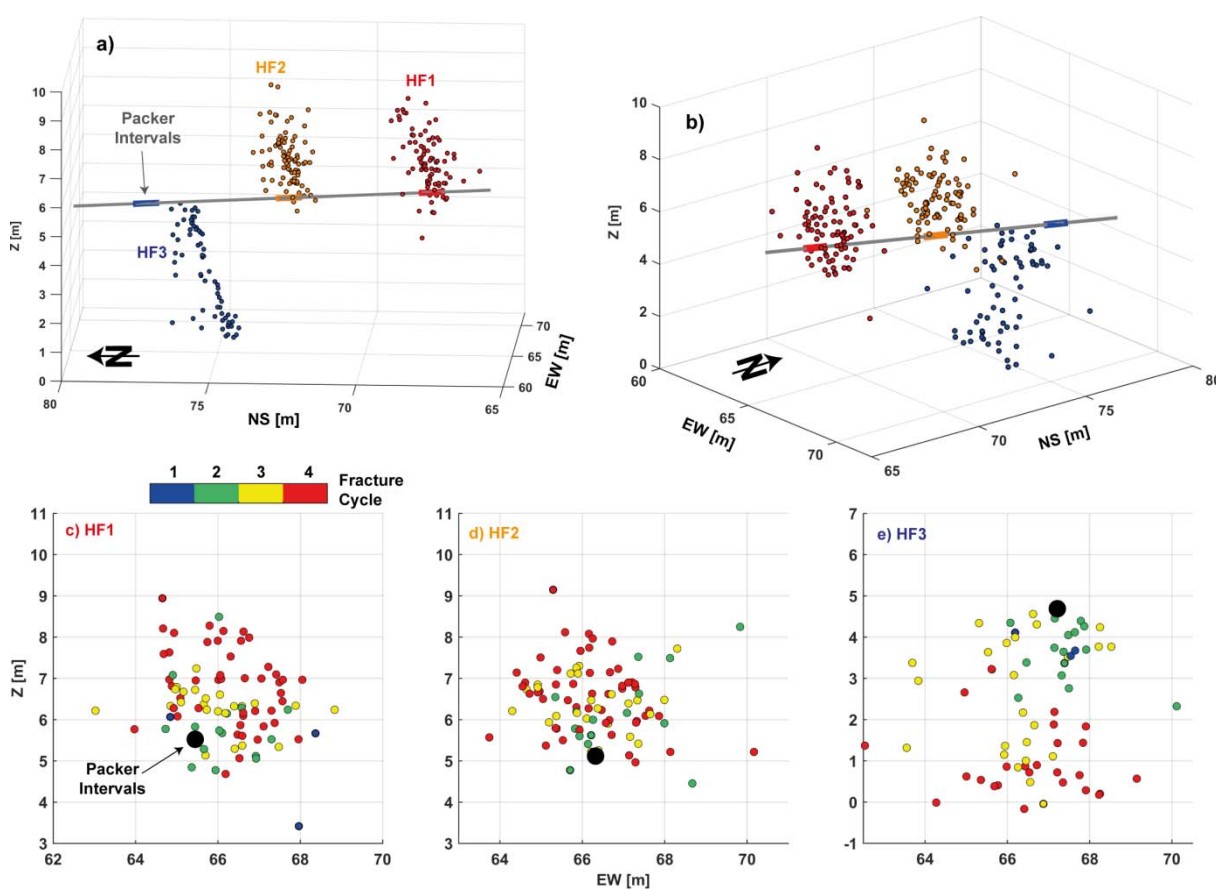

***Figure 5:*** *a) and b) Seismicity clouds of HF1-3 using absolute locations derived from JHD. c) – e) Seismicity clouds (view towards north) with events colored according to the injection cycle during which it occurred.*



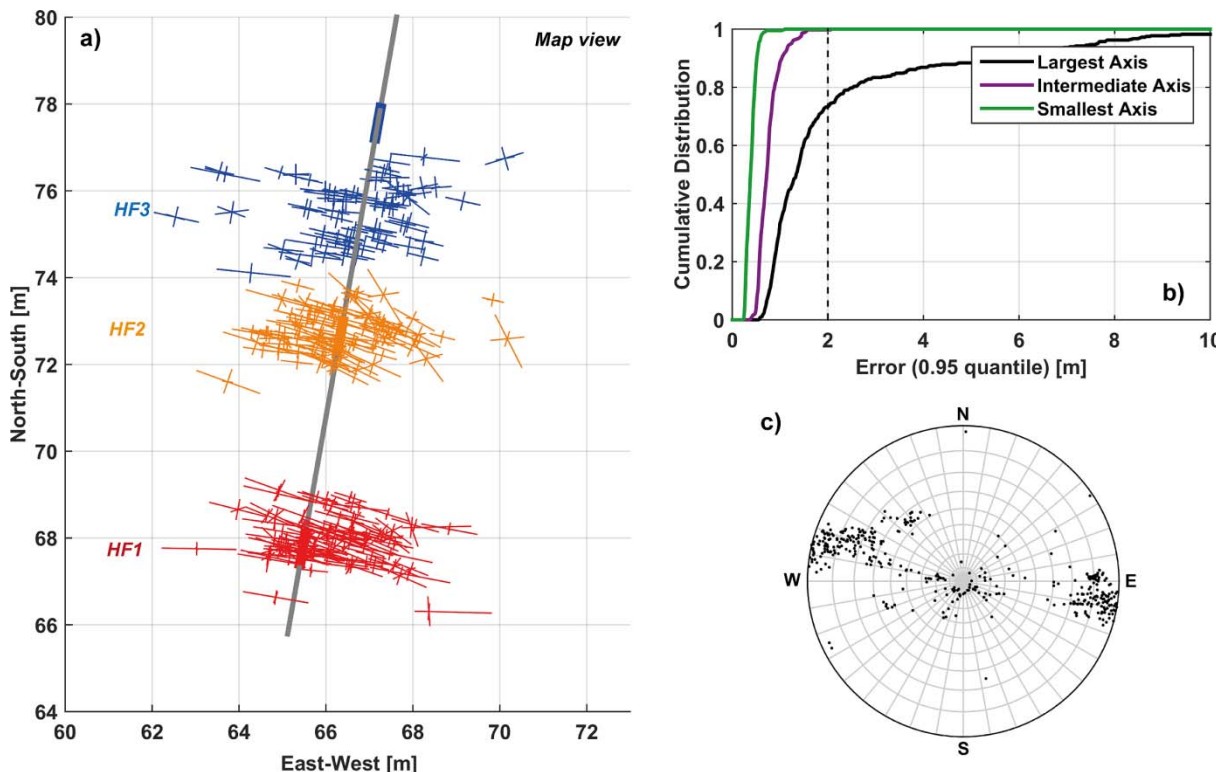

***Figure 6:*** *Error estimates for event locations: a) error ellipsoids shown in map view, b) cumulative distribution of the errors along the three principle axes of the error ellipsoids and c) stereographic projection (lower hemisphere) of the largest error direction. The errors generally are largest in EW direction.*



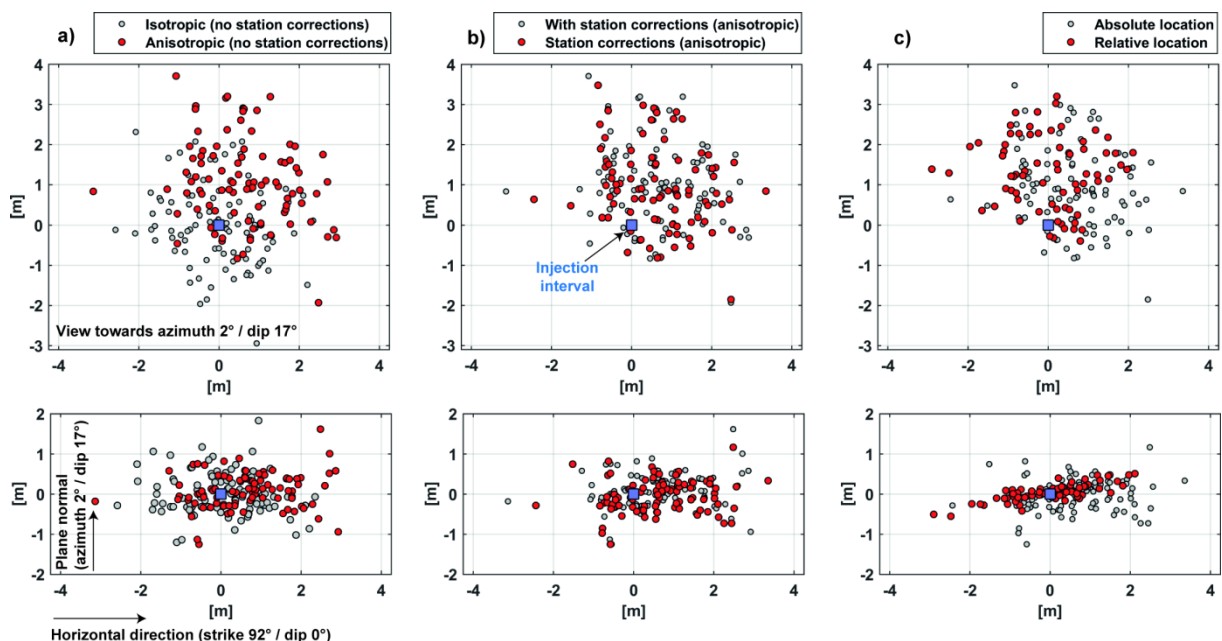


*Figure 7: Impact of anisotropy and station corrections and relative location on source locations. The upper panel is always a projection onto the plane of largest extent of the seismicity clouds. The lower panels are projections perpendicular to the seismicity cloud. a) Isotropic versus anisotropic velocity models with station corrections not included. b) With and without station corrections for the anisotropic velocity model. c) Absolute versus relative locations for the anisotropic velocity model.*



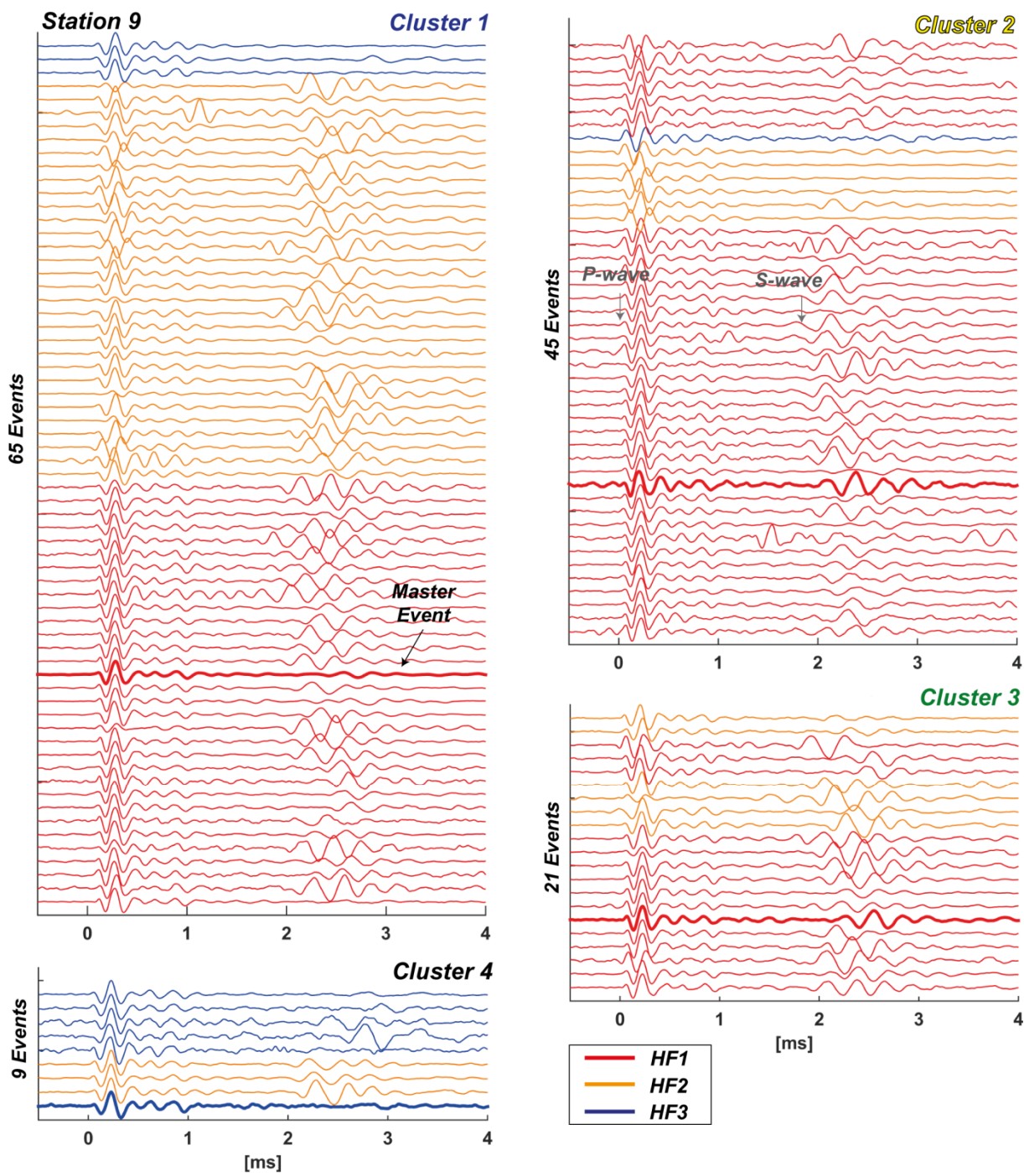

Figure 8: *Selected wave forms events for station S9 and clusters 1-4. Most clusters contain events from several hydraulic fracturing positions.*

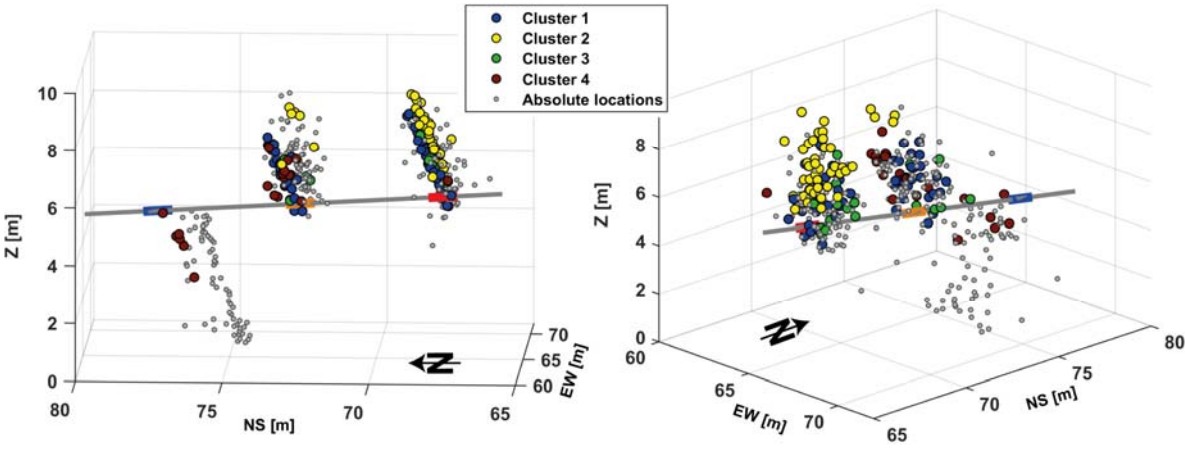


***Figure 9:*** *Relative locations: The hypocentres from relative localization (coloured dots) align along EW planes, with much less scatter than those from absolute localization (grey dots).*



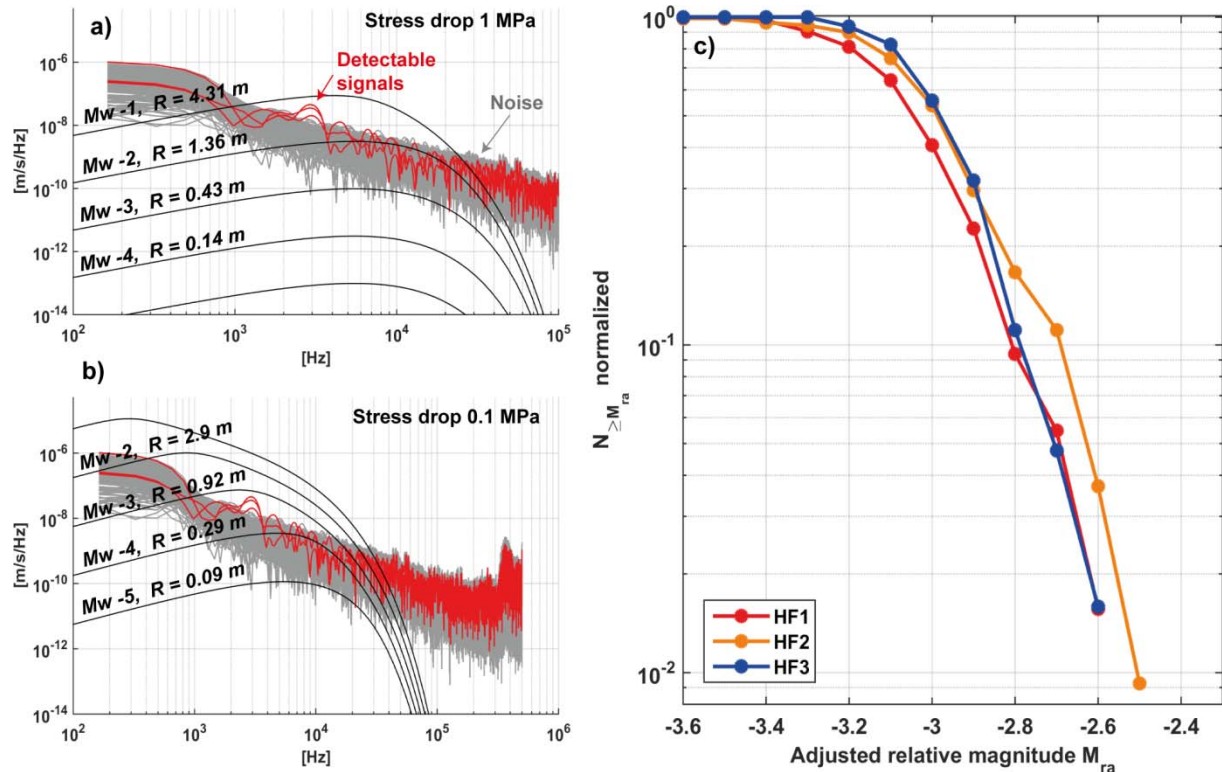

***Figure 10:*** *a) and b) Noise spectra of the accelerometer at sensor position S8 (gray) and the spectra of three events detected at the accelerometer (red) slightly emerging above the noise. Superimposed are theoretical spectra for different magnitudes (Mw -4.0 to -1.0). R denotes the corresponding source radii. The stress drop in a) was chosen to be 1MPa, and in b) it was 0.1 MPa. The detected signals (red) fall in the band between Mw -2.0 and -1.0 for a stress drop of 1 MPa and between Mw -4.0 and -3.0 for stress drop of 0.1 MPa. c) Frequency magnitude distribution of relative adjusted magnitudes $M_{ra}$. These were adjusted so that the maximum magnitude is around $M_{ra}$-2.5 matching a middle value between the approximate maximum magnitude estimates from a) and b).*

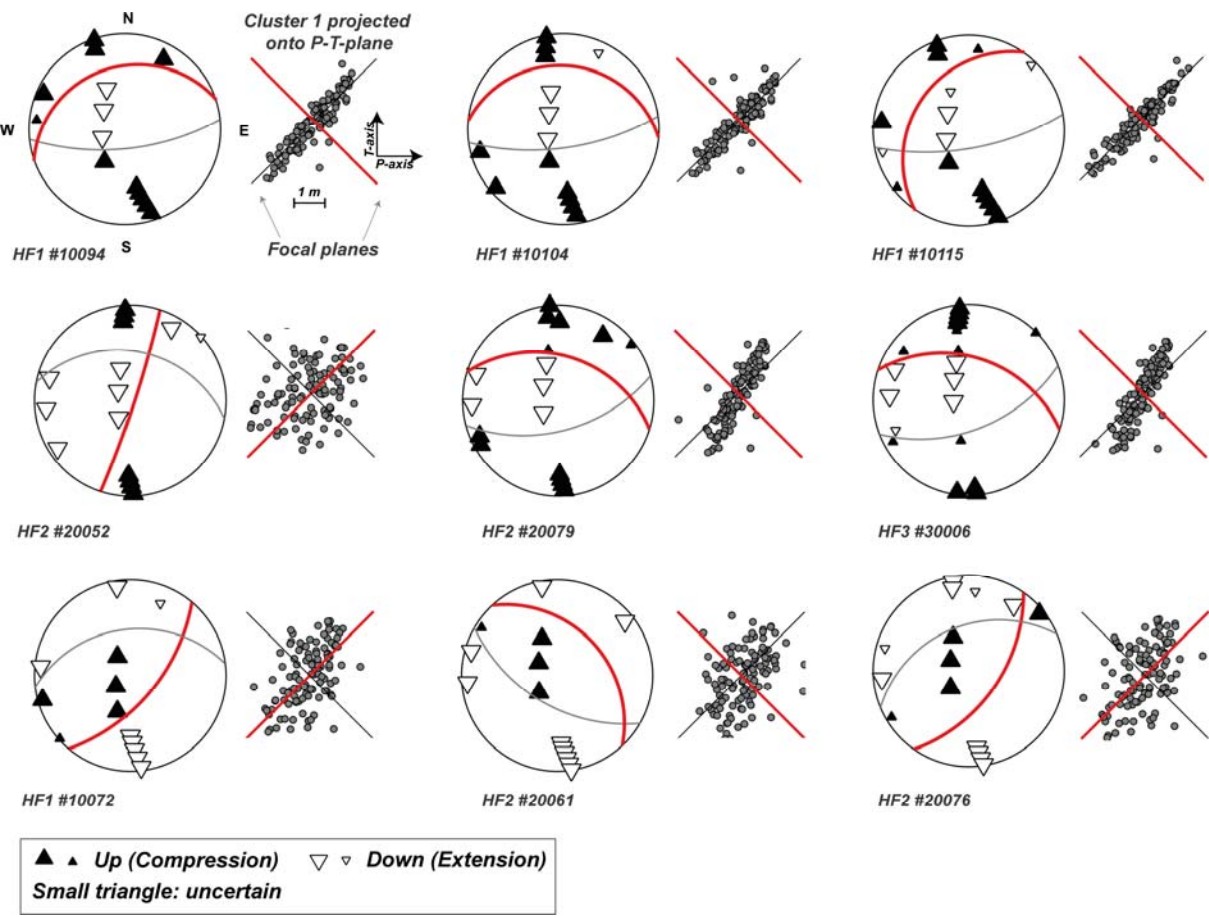

*Figure 11:* *Stereographic plots of focal plane solutions. Next to each focal mechanism is the projection of the seismicity cloud of the corresponding hydrofracture onto a plane defined by the P- and the T-axis where the focal planes appear as a diagonal cross. If the seismicity cloud orientation agrees with one the focal planes, the seismic events group closely around one of the planes. Note that the focal plane, on which stress was resolved in Figure 14, is marked as red line.*

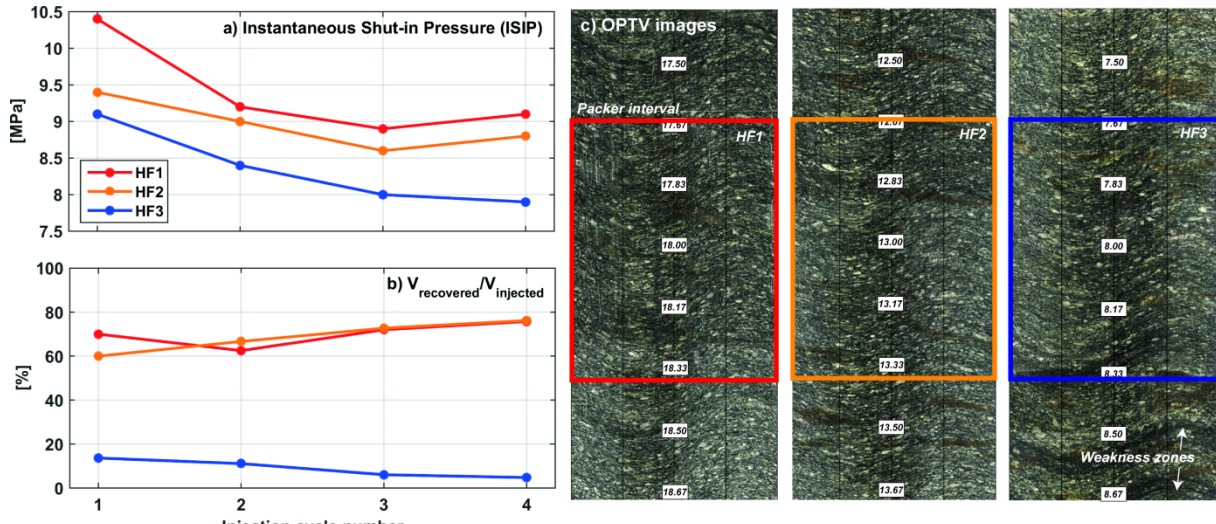

***Figure 12:*** *Hydrofracturing results: a) Development of Instantaneous shut-in pressure (ISIP) and b)*
*relative volume recovery of injected water with cycle for the three hydro-fractures. For HF3, ISIP*
*continues to decrease with each cycle and the recovered volume is very low. c) Optical televiewer*
*images of the three hydrofracturing intervals.*

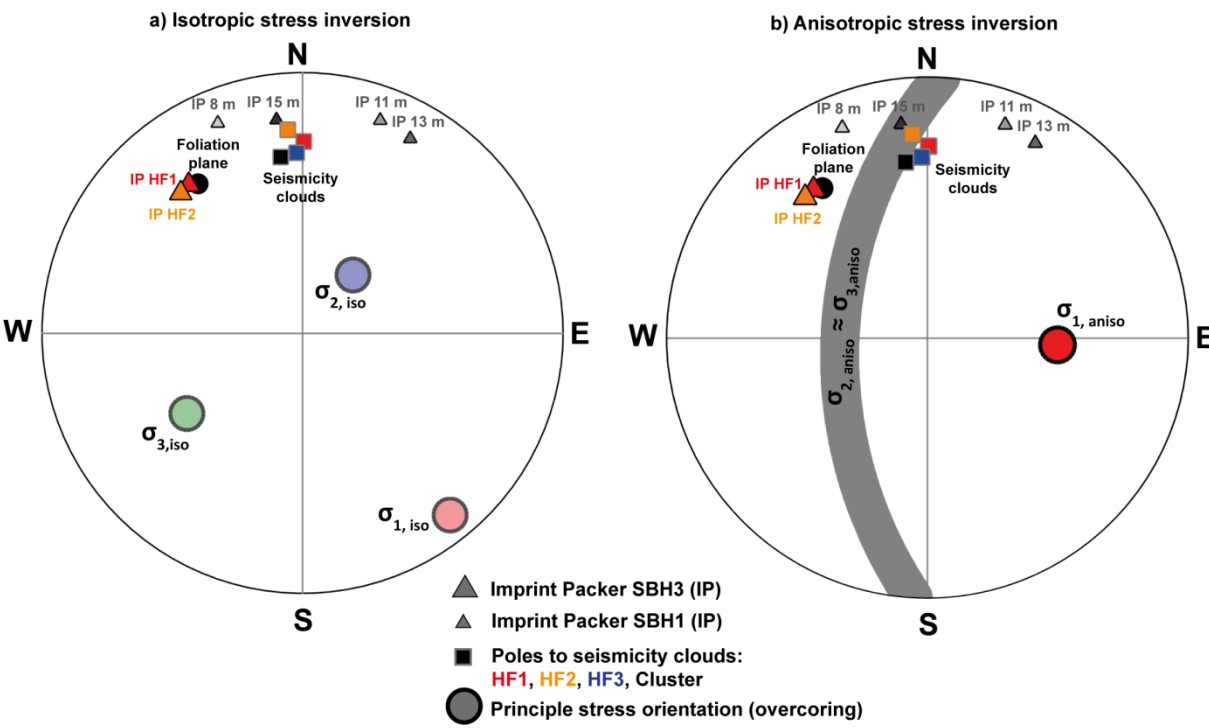

***Figure 13:*** *Comparison of seismicity cloud directions with the foliation plane, fractures mapped on imprint packers (IP) and the principal stress directions from overcoring (σ₁₋₃) with the seismicity clouds. a) For stress inversion of the overcoring assuming isotropic elastic parameters, and b) for transversely isotropic elastic parameters (Krietsch et al., 2017).*

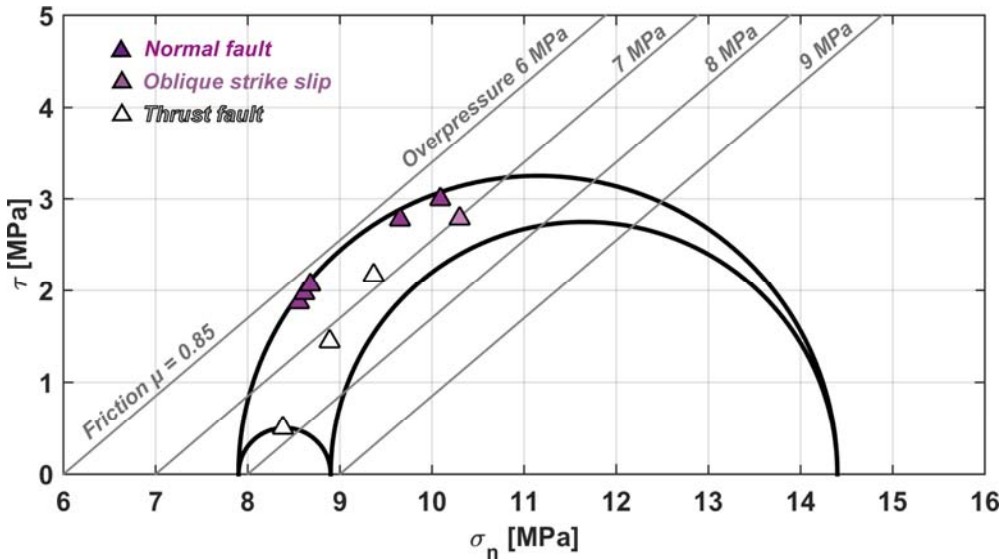

*Figure 14: Mohr-Coulomb diagram representing the stress field estimate by Krietsch et al., (2017) as Mohr circles (including hydrostatic pressure of 0.6 MPa). The failure limits assuming a friction coefficient of 0.85, no cohesion and ovserpressures of 6, 7, 8 and 9 MPa are shown. For the focal mechanisms in Figure 11 the normal stress and shear stress are computed for the  focal plane that requires the smallest overpressure (above hydrostatic) to reach failure. All selected focal planes fail*

*for overpressures of 7 – 9 MPa..The focal planes, for which stresses are represented, are indicated in red in Figure 11.*
