# Peer review of "On the link between stress field and small-scale hydraulic fracture growth in anisotropic rock derived from microseismicity"

_Solid Earth, 2017_

## Referee Comment (RC1) · Anonymous Referee #1 · 2 Aug 2017

This paper on the Grimsel experiment is very important and will be a large contribution to understanding the link between stress and micro/nano seismicity. Just as important are the techniques uses to measure the seismicity , the description of the instrumentation and the bandwidth covered.

An important result is the finding that closer one is to the hydrofractures the better, this will help in designing future experiments to examine all modes of failure

---

## Referee Comment (RC2) · G. Kwiatek (Referee) · 3 Aug 2017

General comments

The paper by Gischig et al. aims at characterizing the local stress field at Grimsel underground rock laboratory during the small-scale hydraulic fracturing experiment. To achieve this task, the Authors combine a variety of direct in-situ stress measurements with seismic data originating from high-sensitivity acquisition system consisting of in-situ acoustic emission (AE) sensors an high-frequency accelerometers.

The manuscript is an example of solid scientific work combining seismic data (here:

[Figure]

extremely small seismic events recorded with the great effort using state-of-the-art high-frequency acquisition system) with geomechanical (local stress measurements), as well as geological data towards a unified, seismo-mechanical characteristics of the local stress field. There is not so many comparable projects, e.g. recent fatigue hydraulic fracturing experiment in ASPO (Zang et al., GJI, 2017), or at even smaller scales in the laboratory, and they all posed a significant challenges in data acquisition and interpretation. As such, the presented work contributes to the scientific progress in understanding the seismic response and stress field evolution in geo-reservoirs subjected to hydraulic fracturing. The applied investigation methods, although not innovative, are absolutely valid and appropriately applied, as well as appropriately referenced whenever necessary. The paper is well written and structured. The quite lengthy introduction could be likely slightly reduced by focusing more on actual topic of the paper (stress field characterization). The number of figures is appropriate. The readability of figures could be improved (I provided some comments later on). As such, I think the manuscript is acceptable for publication after addressing proposed comments.

Specific comments

The Authors put a lot of efforts in constraining the hypocenters of AE activity. However, I am puzzled why they did not use the S-waves to improve the location quality? From Figure and the paper itself it is clear the S-waves were efficiently recorded and they could help to constrain the locations. S-phases have been applied in previous studies using similar acquisition system with success (JAGUARS project, ASPO FHF experiment, see appropriate papers). Could you comment on that and also inform the Readers on your choices?

In introduction (L145-149) I think it would be fair and beneficial to mention in this context a concurrent Aspo FHF experiment (Zang et al., 2017; also: Kwiatek, G., Martínez-Garzón, P., Plenkers, K., Leonhardt, M., Zang, A., Specht, S., Dresen, G., and M. Bohnhoff. Insights into complex sub-decimeter fracturing processes occurring during water-injection experiment at depth in Äspö Hard Rock Laboratory, Sweden, JGR, submitted, or equivalently a similar, but already published contribution at the Schatzalp workshop on Induced Seismicity). The aim of ASPO experiment was to optimize the hydraulic fracturing procedure and limit the occurrence of undesirable LME. Otherwise it seems your introduction is incomplete.

The calculation of average cross-correlation coefficient (L293-294) seems to be not conducted fully appropriately. The ensemble of correlation coefficients should be first transferred to the Fisher's domain (Z-domain) and then averaged. The resulting average can be transferred back to the "regular" domain by the inverse Z-transform (variance stabilizing transformation). Please correct.

Regarding the observations related to shear/tensile events (L390-L392). I would definitely appreciate here more extensive presentation of results in exchange for the pure reference to Eaton's paper. What is the range of ES/EP energies (or S/P amplitudes) you observe? Do the observed values of ES/EP or S/P amp. ratio allow to suggest the existence of tensile components? (cf. discussion in Kwiatek, G. and Y. Ben-Zion (2013). Assessment of P and S wave energy radiated from very small shear-tensile seismic events in a deep South African mine. J. Geophys. Res. 118, 3630-3641, DOI: 10.1002/jgrb.50274.).

In lines L473-L475: Stress rotations due to pressure changes were already reported for geothermal reservoir (Martínez-Garzón, et al., GRL, (2013), DOI: 10.1002/grl.50438) and also confirmed recently by synthetic modellings (see Ziegler et al., Schatzalp 2017). If you want to keep this section (see minor comments), that could benefit to the discussion.

The comparison of stress estimations (~L530-532). I am really puzzled why the Authors did not perform the stress tensor inversion using the polarity data, e.g. following the nonlinear stress tensor inversion (MOTSI Abers/Gephart), and rely their results on the spatial orientation of the fault planes. I believe stress tensor inversion would provide additional information supporting your findings.

[Figure]

I am afraid I am not fully understanding your discussion on mechanisms (L570-580). Looking at your mechanisms the variability of the fault plane solutions does not seem extreme. Why don't you plot a Mohr circle and discuss how (even the composite) fault plane solutions fits to the stres field derived from other measurements. It may be that the fault planes (regardless of whether they are normal, strike slip or thrust) may actually be critically stressed in this complex environment. This was successfully applied for analyzing ASPO data (see Kwiatek et al., 2017, Schatzalp proceedings).

Minor comments

L65 I understand that the volume is related to small-scale hydrofracturing projects. However, it would be good to write it down here once again explicitly to distinguish it from larger-scale industrial projects

L73 Naming convention for "microseismic" is not consistent. You either use micro-seismic or microseismic. Please unify.

L83 Manthei

L124 Zang et al., 2017 (and everywhere else, the appropriate reference is: Zang, A., Stephansson, O., Stenberg, L., Plenkers, K., Specht, S., Milkereit, K., Schill, E., Kwiatek, G., Dresen, G., Zimmermann, G., Dahm, T., and M. Weber (2017), Hydraulic fracture monitoring in hard rock at 410m depth with an advanced fluid-injection protocol and extensive sensor array, Geophys. J. Int., 208(2), 790–813, DOI: 10.1093/gji/ggw430)

L215 The Wilcoxon does not seem to work very efficiently above 17kHz (cf. appendix in Kwiatek et al., 2011) and our experience was that the transfer function was not flat above this range.

L217 You use "piezo-sensor", however the well-defined term in the field (cf. Kwiatek et al. 2011 and other follow-ups) is the "(in-situ) acoustic emission sensor" or simply "acoustic emission sensor".

L226 SBH-1

L229-230 This is in surprising agreement with what was observed at ASPO FHF (reference?)

L244 Correct the reference

L251 How the P-wave velocity was estimated (any details, reference, variability?)

L279-L284 It is not clear what this procedure reflects. Are you concerned about velocity mismodelling, or P-wave mispicking. Please clarify.

L286 Is it 95% confidence interval?

L323-325 This is similar to what has been observed at ASPO, please refer.

L357-L385 The sentence starting with "Clear rings..." is not understandable, please rephrase/modify/extend it.

L380 all 112

L411 $\pi f_0$ L412 P-wave

L413 What is the range of dominant frequencies you have and how this correspond to the resonant frequencies of the AE sensors? Does it not lead to overestimation of magnitude, as you enhance the resonances at high frequencies due to Q correction?

L420 Unfortunately to the project, this suggest putting accelerometer sensors on tunnel walls does not allow to effectively record such small seismic events despite of reasonable source-receiver distances.

L425 This part is hard to understand. Please refer to exact figures. The caption of figure 8 is poor - it is hard to understand differences between a) and b) (stress drop...), Please explain all the details of the figure in the caption!!!

L426 Guess there is a typo here. I have no idea how you calculated so small source radii for the assumed stress drops of 1MPa. Using Eshelby's formula bring me to way

larger values of the order of meters (0.8-2.5m i.e. 80 to 260 cm, not 8 to 25 cm as you write) for the stress drop of 1MPa. Anyway, calculations for 0.1MPa seems to be fine. It is also more reasonable assumption regarding the frequencies and expected source radii as well as the low expected confining pressure (stress drop seems to be dependent on average stress)

L440 What is the b-value? Is it comparable with what is observed for other fracking project (high values)?

L453 Isn't it simply the composite fault plane solution?

L460-L461 I think it is over-intepretation. I guess if you stack multiple double-couple sources with even slightly different focal mechanisms you may achieve spurious non-DC components (e.g. Frohlich, 1994). Therefore, i am not sure if your reasoning do not go to far. I guess waveform stacking is not the way you can get a reasonable message with respect to the mechanism of faulting (shear/tensile). Please reduce L455-L465 to solid observables and reduce the discussion.

L467-L475 This paragraph should go to discussion, if you are concerned about the (apparent) lack of non-DC events. I am not concerned and I do support your findings, though, and believe your results are solid without too much speculations on why you don't observe tensile openings. As you write, the tensile openings are energetically ineffective, and likely limited to the very small earthquakes that likely cannot be handled properly by AE system.

L493 Nice observation

L500 Some crossed-out words

L530-532 Some more crossed-out words

L559 Stress magnitudes and stress field should be presented earlier in the manuscript (just before section 3 i suggest)

L594 "readily be applied to failure at such scales"? Is that what really want to write?

Figure 2 "d)" is misplaced. The idea of coloring per HF is a good one, but the problem is you re-use these colors for painting other curves (e.g. Injection pressure and rate in Figure 2). Maybe it is possible to take advantage of other colors or patterns (e.g. dashed lines) in this and other figures while you NOT refer to the particular stimulation. Finally I suggest to repeat HF# label in each panel and also paint dots in g,h,i with appropriate color reflecting HF.

Figure 4. Hard to read... Why not to put simply a 2D version of this plot (e.g. view from top)?

Figure 6. Mark HF1 HF2 and HF3 labels in a). I suggest to change colors in b) to not confuse reader with what is in the subplot a)

Figure 10. Please rewrite the whole caption. The content of the figure is quite badly explained. What is R? Please enhance stress drop messages, or just explain in caption what is the difference between a) and b) subplot.

Figure 11. Remove "for a few events". I suggest to put shading to distinguing thrust from normal focal mechanisms more easily.

---

## Author Comment (AC1) · 12 Sep 2017

**Response to reviewers**

**Reviewer 1**

*We thank to reviewer 1 for the positive comments on our manuscript.*

**Reviewer 2 (G. Kwiatek)**

*We would like to thank to G. Kwiatek for his thorough review, which we led to great improvement of the manuscript. We were able to address almost all of his comment as detailed below.*

The Authors put a lot of efforts in constraining the hypocenters of AE activity. However, I am puzzled why they did not use the S-waves to improve the location quality? From Figure and the paper itself it is clear the S-waves were efficiently recorded and they could help to constrain the locations. S-phases have been applied in previous studies using similar acquisition system with success (JAGUARS project, ASPO FHF experiment, see appropriate papers). Could you comment on that and also inform the Readers on your choices?

*Although in many events the S-waves are clearly discernible, we found that the onset times of only a few of them could be accurately picked, while most of them are associated with considerable uncertainty. Furthermore, the S-wave velocity of the rock mass is not well described in the literature, and our S-wave arrival times are too few to determine a reliable S-wave velocity model. For these reasons, and the fact that the locations are already well determined with P-wave arrivals only, we decided that the effort to pick and use S-waves arrival is more trouble that it is worth.*

In introduction (L145-149) I think it would be fair and beneficial to mention in this context a concurrent Aspo FHF experiment (Zang et al., 2017; also: Kwiatek, G., Martínez- Garzón, P., Plenkers, K., Leonhardt, M., Zang, A., Specht, S., Dresen, G., and M. Bohnhoff. Insights into complex sub-decimeter fracturing processes occurring during water-injection experiment at depth in Äspö Hard Rock Laboratory, Sweden, JGR, submitted, or equivalently a similar, but already published contribution at the Schatzalp workshop on Induced Seismicity). The aim of ASPO experiment was to optimize the hydraulic fracturing procedure and limit the occurrence of undesirable LME. Otherwise it seems your introduction is incomplete.

*We included the proposed references in the introduction.*

The calculation of average cross-correlation coefficient (L293-294) seems to be not conducted fully appropriately. The ensemble of correlation coefficients should be first transferred to the Fisher's domain (Z-domain) and then averaged. The resulting average can be transferred back to the "regular" domain by the inverse Z-transform (variance stabilizing transformation). Please correct.

*We included the Fisher transformation in our cluster analysis procedure (also mentioned in now in the manuscript) and found comparable results to the previous analysis without the Fisher transformation. We updated Figure 8 and 9 with the new results and adjusted the text accordingly. As the events for which focal mechanisms were computed do not fall into the same clusters as before, we removed the grouping into events belonging to different clusters. The results and interpretations remain the same.*

Regarding the observations related to shear/tensile events (L390-L392). I would definitely appreciate here more extensive presentation of results in exchange for the pure reference to Eaton's paper. What is the range of ES/EP energies (or S/P amplitudes) you observe? Do the observed values of ES/EP or S/P amp. ratio allow to suggest the existence of tensile components? (cf. discussion in Kwiatek, G. and Y. Ben-Zion (2013). Assessment of P and S wave energy radiated from very small shear-tensile

seismic events in a deep South African mine. J. Geophys. Res. 118, 3630-3641, DOI: 10.1002/jgrb.50274.).

*We expanded the short discussion on possible presence of tensile source components by giving the range of observed S/P-wave amplitude ratios, and mention that low values may point to a tensile component, as found by Kwiatek and Ben-Zion (2013) (added as reference).*

In lines L473-L475: Stress rotations due to pressure changes were already reported for geothermal reservoir (Martínez-Garzón, et al., GRL, (2013), DOI: 10.1002/grl.50438) and also confirmed recently by synthetic modellings (see Ziegler et al., Schatzalp 2017). If you want to keep this section (see minor comments), that could benefit to the discussion.

*We moved this section to the discussion section and added the suggested reference Martínez-Garzón, et al., (2013). We also added the earlier paper on stress perturbations due to hydraulic fracturing by Warpinski & Branagan, Altered stress fracturing, JPT,1998.*

The comparison of stress estimations (_L530-532). I am really puzzled why the Authors did not perform the stress tensor inversion using the polarity data, e.g. following the nonlinear stress tensor inversion (MOTSI Abers/Gephart), and rely their results on the spatial orientation of the fault planes. I believe stress tensor inversion would provide additional information supporting your findings.

*Stress inversion from polarization data is not the primary goal of this paper and may in fact produce very misleading results. Instead, we strive to obtain reliable orientations of the seismicity clouds that provide an indication to the σ3 direction, and to compare these to the results of other stress measurements. Clearly, the found focal mechanisms do not match well to the the stress field orientation estimates from both the seismicity clouds and the overcoring measurements. As discussed later, the focal mechanisms may indicate the stress perturbations around the propagating fracture, which in our case would invalidate the use of polarization data to infer stress orientations.*

I am afraid I am not fully understanding your discussion on mechanisms (L570-580 ). Looking at your mechanisms the variability of the fault plane solutions does not seem extreme. Why don't you plot a Mohr circle and discuss how (even the composite) fault plane solutions fits to the stres field derived from other measurements. It may be that the fault planes (regardless of whether they are normal, strike slip or thrust) may actually be critically stressed in this complex environment. This was successfully applied for analyzing ASPO data (see Kwiatek et al., 2017, Schatzalp proceedings).

*By plotting the stress conditions along the focal planes in a Mohr-Coulomb diagram, we indeed find that pressure leak-off is sufficient to explain the different focal mechanisms. We added such a figure to the text and adjusted the discussion on the topic accordingly. Thanks for the great idea.*

Minor comments

L65 I understand that the volume is related to small-scale hydrofracturing projects. However, it would be good to write it down here once again explicitly to distinguish it from larger-scale industrial projects

*Corrected.*

L73 Naming convention for "microseismic" is not consistent. You either use microseismic or microseismic. Please unify.

*We replace micro-seismic by microseismic throughout.*

L83 Manthei

*Corrected.*

L124 Zang et al., 2017 (and everywhere else, the appropriate reference is: Zang, A., Stephansson, O., Stenberg, L., Plenkers, K., Specht, S., Milkereit, K., Schill, E., Kwiatek, G., Dresen, G., Zimmermann, G., Dahm, T., and M. Weber (2017), Hydraulic fracture monitoring in hard rock at 410m depth with an advanced fluidinjection protocol and extensive sensor array, Geophys. J. Int., 208(2), 790–813, DOI:10.1093/gji/ggw430)

*Corrected.*

L215 The Wilcoxon does not seem to work very efficiently above 17kHz (cf. appendix in Kwiatek et al., 2011) and our experience was that the transfer function was not flat above this range.

*Corrected.*

L217 You use "piezo-sensor", however the well-defined term in the field (cf. Kwiatek et al. 2011 and other follow-ups) is the "(in-situ) acoustic emission sensor" or simply "acoustic emission sensor".

*We prefer to keep this term (that is also used in other articles) as we feel it is less ambiguous. Acoustic emission in our understanding refers to a target frequency band. Thus, the term 'acoustic emission sensor' does not say anything about the sensor technology; both piezosensors and accelerometers may be called 'acoustic emission sensors' as they can cover the acoustic frequency range.*

L226 SBH-1

*Corrected throughout.*

L229-230 This is in surprising agreement with what was observed at ASPO FHF (reference?)

*I did not find such an observations stated by Zang et al., 2017 or any other ASPO reference.*

L244 Correct the reference

*Corrected.*

L251 How the P-wave velocity was estimated (any details, reference, variability?)

*We already describe that the P-wave velocity model is estimated using a grid-search approach minimzing the differences between observed and predicted P-wave arrival time. We feel the information given is already appropriate*

L279-L284 It is not clear what this procedure reflects. Are you concerned about velocity mismodelling, or P-wave mispicking. Please clarify.

*We now clearly indicate in the manuscript that we are concerned with P-wave arrival time uncertainties.*

L286 Is it 95% confidence interval?

*Yes. We added this in the manuscript.*

L323-325 This is similar to what has been observed at ASPO, please refer.

*We added reference to Zang et al., 2017.*

L357-L385 The sentence starting with "Clear rings..." is not understandable, please rephrase/modify/extend it.

*We formulate this statement more clearly now.*

L380 all 112

*Corrected.*

L411 npi f_0

*Corrected.*

L412 P-wave

*Corrected.*

L413 What is the range of dominant frequencies you have and how this correspond to the resonant frequencies of the AE sensors? Does it not lead to overestimation of magnitude, as you enhance the resonances at high frequencies due to Q correction?

*The range of dominant frequencies is between 1 – 10 kHz as not indicated in the manuscript. To avoid artificial enhancement at (currently unknown) sensor resonance frequencies, we follow the recommendation of Zang et al., (2017) and filter data between a narrow band-pass filter between 3 and 7 kHz and use nominal frequency $f_0$ = 5 kHz to correct for attenuation.*

L420 Unfortunately to the project, this suggest putting accelerometer sensors on tunnel walls does not allow to effectively record such small seismic events despite of reasonable source-receiver distances.

*Although we agree that this might be the case, it does not naturally conclude from our study, as we did not compare borehole versus tunnel wall installation of accelerometers. Thus, we refrain from commenting on this in the manuscript.*

L425 This part is hard to understand. Please refer to exact figures. The caption of figure 10 is poor - it is hard to understand differences between a) and b) (stress drop...), Please explain all the details of the figure in the caption!!!

*We expanded the caption with details so that figure 10 and caption become self-explanatory. We also improved the details of this computation in the manuscript.*

L426 Guess there is a typo here. I have no idea how you calculated so small source radii for the assumed stress drops of 1MPa. Using Eshelby's formula bring me to way larger values of the order of meters (0.8-2.5m i.e. 80 to 260 cm, not 8 to 25 cm as you write) for the stress drop of 1MPa. Anyway, calculations for 0.1MPa seems to be fine. It is also more reasonable assumption regarding the frequencies and expected source radii as well as the low expected confining pressure (stress drop seems to be dependent on average stress)

*Indeed, we must have made an error in the calculations. The source radii lie between 1.4 and 4.3 meter for stress drop of 1 MPa and 0.3 – 0.9 m for 0.1 MPa stress drop. We changed the numbers in the text and figure 10 as well as the corresponding discussion accordingly.*

L440 What is the b-value? Is it comparable with what is observed for other fracking project (high values)?

*Due to the limited number of events with magnitudes and associated high uncertainty in b-value computation, we decided to report only report a robust lower bound for b-values that is estimated to be >2.*

L453 Isn't it simply the composite fault plane solution?

*Due this and the following comment we decided to remove these focal mechanisms based on stacked waveform polarizations.*

L460-L461 I think it is over-intepretation. I guess if you stack multiple double-couple sources with even slightly different focal mechanisms you may achieve spurious non-DC components (e.g. Frohlich, 1994). Therefore, i am not sure if your reasoning do not go to far. I guess waveform stacking is not the way you can get a reasonable message with respect to the mechanism of faulting (shear/tensile). Please reduce L455-L465 to solid observables and reduce the discussion.

*We agree with the reviewer that this is possibly beyond any reasonable interpretation, as it is not clear what such 'stacked focal mechanisms' represent. We therefore remove them from Figure 11 and the corresponding text in the manuscript, and thank the reviewer for pointing out a reference that has already considered the problem.*

L467-L475 This paragraph should go to discussion, if you are concerned about the (apparent) lack of non-DC events. I am not concerned and I do support your findings, though, and believe your results are solid without too much speculations on why you don't observe tensile openings. As you write, the tensile openings are energetically ineffective, and likely limited to the very small earthquakes that likely cannot be handled properly by AE system.

*We agree and moved the paragraph to the discussion, where parts of it are repeated anyway (we kept the first sentence as it goes well with the end of the previous paragraph). We also included the reference to Martinez-Garzon et al., (2013) that was recommended earlier.*

L493 Nice observation

*Thanks!*

L500 Some crossed-out words

*Corrected.*

L530-532 Some more crossed-out words

*Corrected.*

L559 Stress magnitudes and stress field should be presented earlier in the manuscript
(just before section 3 i suggest)

*We moved the table with the stress information to the end of the section on stress measurement.*

L594 "readily be applied to failure at such scales"? Is that what really want to write?

*We changed the sentence to 'can be applied to seismicity at such scales'*

Figure 2 "d)" is misplaced. The idea of coloring per HF is a good one, but the problem is you re-use these colors for painting other curves (e.g. Injection pressure and rate in Figure 2). Maybe it is possible to take advantage of other colors or patterns (e.g. dashed lines) in this and other figures while you NOT refer to the particular stimulation. Finally I suggest to repeat HF# label in each panel and also paint dots in g,h,i with appropriate color reflecting HF.

*We improved the figure as suggested by the reviewer.*

Figure 4. Hard to read... Why not to put simply a 2D version of this plot (e.g. view from top)?

*A 2D plot would conceal the borehole seismic array, which nicely shows the impact of the anisotropy on station corrections. Indeed the 3D view is difficult to read, however, we could not find a better view angle.*

Figure 6. Mark HF1 HF2 and HF3 labels in a). I suggest to change colors in b) to not confuse reader with what is in the subplot a)

*Changed as suggested.*

Figure 10. Please rewrite the whole caption. The content of the figure is quite badly explained. What is R? Please enhance stress drop messages, or just explain in caption what is the difference between a) and b) subplot.

*We changed the figure and caption accordingly and added more information (see comment further up).*

Figure 11. Remove "for a few events". I suggest to put shading to distinguing thrust from normal focal mechanisms more easily.

*Caption changed as suggested. We feel the focal mechanisms are more clear without shading.*

---

## Author Comment (AC2) · 4 Oct 2017

[revised manuscript text omitted]

~~For Clusters 1 and 3, we also derived polarizations by stacking the wave forms of all cluster events. If the events all have the same mechanism, waveform stacking would show reliable polarizations. These polarizations are shown in Figure 11, where we consider the different source locations within the cluster that give rise to slight change in radiation direction. Although the polarization pattern is comparable to the patterns from single events, it was not possible to fit a double-couple mechanism. Currently, we cannot explain the discrepancy. Possibly the mechanisms of the events within the cluster vary (as the variable S/P-wave amplitude ratios also indicate; Figure 8), and thus waveform stacking does not give meaningful polarizations. Another reason may be that there are components of non-double-couple mechanisms affecting the polarization pattern.alsoobserved for Clusters 1 and 3,~~ contradicts the stress field observations. As described in Section 5.2, the maximum principal stress $\sigma_1$ and $\sigma_3$ are sub-horizontal and $\sigma_2$ and $\sigma_3$ are very close in magnitude suggesting that a thrust fault to strike-slip mechanism is expected. Possibly, a component of volumetric expansion or a compensated linear vector dipole (CLVD) mechanism modifies the pure double-couple mechanisms (Vavrycuk, 2011). Volumetric expansion would be consistent with growth of a tensile fracture driven by fluid injection. The double-couple mechanisms found here are in agreement with many studies that showed that seismicity associated with HF have double-couple sources (Chitrala et al., 2013; Dahm et al., 1999; Nolan-Hoeksema & Ruff, 2001; Ohtsu, 1991). ~~The events are thought to be induced either by pressure propagating into the small pre-existing fractures adjacent to growing HF (e.g. Rutledge and Phillips, 2004) or by stress changes induced around the propagating HF (as suggested by Nolen-Hoeksema & Ruff, 2001). In the latter case, it is expected that the stress field locally rotates and that deviations from the ambient stress field reflected in the focal mechanisms occur. Future analysis of seismicity of similar experiment performed at the GTS may give more insights into this.~~

[revised manuscript text omitted]

$\dfrac{\partial \theta}{\partial s^{x}} = -\dfrac{1}{\sqrt{1-\Gamma^2}}\dfrac{\partial \Gamma}{\partial s^{x}}$

$= -\dfrac{1}{\sqrt{1-\Gamma^2}}\dfrac{\partial}{\partial s^{x}}\left( \cos\phi_{inc}^{sym}\cos\phi_{azi}^{sym}\dfrac{r_i^x-s^x}{l_i} + \cos\phi_{inc}^{sym}\sin\phi_{azi}^{sym}\dfrac{r_i^y-s^y}{l_i} + \sin\phi_{inc}^{sym}\dfrac{r_i^z-s^z}{l_i} \right)$

(22) $\quad = \dfrac{1}{\sqrt{1-\Gamma^2}}\left( \begin{aligned} &\dfrac{\cos\phi_{inc}^{sym}\cos\phi_{azi}^{sym}}{l_i} + \\ &\left\{ \cos\phi_{inc}^{sym}\cos\phi_{azi}^{sym}\dfrac{r_i^x-s^x}{l_i^2} + \cos\phi_{inc}^{sym}\sin\phi_{azi}^{sym}\dfrac{r_i^y-s^y}{l_i^2} + \sin\phi_{inc}^{sym}\dfrac{r_i^z-s^z}{l_i^2} \right\}\dfrac{\partial l_i}{\partial s^x} \end{aligned} \right)$

$\quad = \dfrac{1}{\sqrt{1-\Gamma^2}}\left( \dfrac{\cos\phi_{inc}^{sym}\cos\phi_{azi}^{sym}}{l_i} + \dfrac{\Gamma}{l_i}\dfrac{\partial l_i}{\partial s^x} \right)$

$\quad = \dfrac{1}{l_i\sqrt{1-\Gamma^2}}\left( \cos\phi_{inc}^{sym}\cos\phi_{azi}^{sym} - \Gamma\cos\alpha_i\cos\beta_i \right)$

Similarly, we find

(23) $\quad \dfrac{\partial \theta}{\partial s^{y}} = \dfrac{1}{l_i\sqrt{1-\Gamma^2}}\left( \cos\phi_{inc}^{sym}\sin\phi_{azi}^{sym} - \Gamma\sin\alpha_i\cos\beta_i \right),$

735 and

(24) $\quad \dfrac{\partial \theta}{\partial s^{z}} = \dfrac{1}{l_i\sqrt{1-\Gamma^2}}\left( \sin\phi_{inc}^{sym} - \Gamma\sin\beta_i \right)$

[revised manuscript text omitted]

---

## Referee Comment (RC3) · E Caffagni (Referee) · 14 Nov 2017

**Revision**

**On the link between stress field and small-scale hydraulic fracture growth in anisotropic rock derived from microseismicity**

Gischig, Valentin et al. 2017. Solid Earth Discussions (EGU)

I found this manuscript of great interest, and am glad to see practical demonstration from the ETH group of what promised this year at the Induced seismicity meeting in Davos.
The title matches perfectly the topic developed in the manuscript.
The authors refer to a wide range of references. The experiment set-up, purposes and results are described in a clear way. Results appear to be interesting, and might help to understand rock-breaking processes during geothermal stimulations in Switzerland.

I definitely recommend this work as a publication for the EGU journal. No doubt.
However, from MINOR TO MODERATE REVISIONS are nevertheless needed.

I wish my best, and proficient researches to the ETH group.

Enrico Caffagni

Following some major concerns I have found.

1. The Abstract is quite long.. in comparison to standard EGU Discussions abstracts.

2. The Discussion can be shortened, avoiding repetitions of concepts.

3. Some references are missing or written in an incorrect way.

4. Table 2 is present, but never mentioned in the text.

5. Figure 11 appears with overlapped labels.

6. I feel that a bit of clarification is needed when the authors described the "Microseismic monitoring". I think it is more a question of terminology. The classical induced microseismic events range in frequency between 80 to 300-400 Hz. No more. Sure that the space-time scale is different. Yet you measure events at magnitude -2.5 for instance at KHz. Should we define these as "acoustic emissions"? You actually mention, page 6 line 208-209, "similar to those commonly used in laboratory acoustic emission". For coherency, you should continue to call such events as acoustic emissions, throughout the paper.

7. I am not sure that your stress characterization study is in the 'far-field' domain, page 6 line 184, at least if this is only due to the "several tens of meter away from any fault". What you are doing, I believe, instead is a characterization of the "local" stress field. Which is more useful.

8. It would benefit your paper to include, at a qualitative way, one or two sentences on the topic: what drives the initiation of an hydrofracture, and "when" or "why" it decides to stop? I suggest to refer to Dahm et al. (JGR, 2010), who discuss the effect of the pore pressure gradient, or pressure gradient, which in your experiment might be also responsible of fracture plane deviation or the trend in asymmetrical growth, page 16 line 544.
Better. Your pressure gradient might drive the fracture, but when this starts decreasing, well, your fracture "feels" the local stress field then re-orients itself naturally.
Another plausible explanation of fracture plane rotation is a high resilient tectonic stress, see Cooke et al. (2016, TLE).
(it is just an attempt of clarification; there are better interpretation; work on that)

9. I honestly do not see a strong connection between polarization through stacking and focal mechanisms. In Figure 11 in a few beach-balls a few events appear in different color (black and white) yet in the same section of the ball. This seems to be contradictory.. Actually, by itself, the stacking operations is for enhancing the signal arrival, not for polarization; unless one projects first the traces into already obtained polarization vectors then perform the stacking. By such methods one can locate microseismic events (see Caffagni et al. 2016, GJI).
Beside that, the lack in the majority of strike-slip events or a combination of different source mechanisms seems to be a kind of constant in induced seismicity. You can see and refer for instance to Baisch et al. (2015, BSSA; Figure 12).

10. The authors need to be careful when they declare "the observation of DC components..exclude 1" page 17 line 577-578. No, I do not agree. A tensile source mechanism has a DC component as well. In addition Figure 8 reproduces what to me is a tensile event, Cluster 2 (max P wave is bigger than S wave). If I had to sort hundreds of events automatically and visually, I would classify that event as a tensile one.

11. The authors mention in the Appendix, page 20 line 652, "a solid angle is also referred to as the take-off-angle". The classical take-off-angle is not defined as a solid angle, see Stein and Wysession page 222.

12. More than else, since you as ETH group are currently leading the laboratory experimental research of hydraulic fracturing in Europe, it would benefit to develop on the topic: What's for?
Switzerland might be soon venue of massive usage of geothermal exploitation. What can we learn at macro-scale from such experiments at micro-scale? Can just we simply "upscale" our results in cases of real large-scale stimulations? Perhaps the answer is yes..Injection values in pressure are much higher though and perhaps the effect of small-scale rock anisotropy might vanishes in comparison to big deformations due to pore-elastic effects or fluid diffusion or fracturing.
The paper you want to publish is called "EGU Discussion". It would be appropriate if you could "unbalance" yourself and make some qualitative or even quantitative declarations on future perspectives from such mini-scale hydro-fracturing experiments.

In the following you will find my minor concerns. Please, do not reply in your response file to all of them. But make sure to read them and revise. I have also added additional important concepts missing.

**Minor concerns**

**Abstract**

Pg. 1

Line 15: "transverse" to what? To the radial? Usually P-waves propagates in the radial direction

Line 20: " from the overcoring stress. An anisotropic elastic model.."

Line 21 "sigma1 is significantly.." to Line 24 "the north". Is not clear and I would simply remove it, to shorten the abstract

**Introduction**

Pg. 2

Line 37: "Hydraulic fracturing or hydrofracking (HF)"

Line 37: Rephrase "Hydraulic fracturing induces artificial fracture networks in a rock mass by high-pressure fluid-injection. It has become an essential technique…, for instance to enhance the permeability…, and increase… . HF should not be confused with hydroshearing (HS). HS is a method of rock.. that uses fluid-injection….promoting shear failure, attendant dilation of pre-existing fractures, and fault slippage…"

Line 45: "criticality of the discontinuity sets" What do you mean? It is not clear

Line 47: "HS has been often exploited in… . HF small volume has been also utilized in stress.."
I would replace the "deep geothermal projects" with the EGS, the "Enhanced Geothermal systems"

Line 52: "etc; see Zang and...2010"

Line 52: "To better constrain the stress field...and sections where no pre-existing fractures are present"

Line 55: "water is injected at a constant rate until the ..down, initiating a fracture at the borehole wall...sub parallel to the principal…., and significant deviations are not expected due to the tensile strength anisotropy.."
Is this what you mean with "no complications from tensile..." If not, please provide explanations.

Line 57: ", then high-pressure fluid injected will tend to initiate an axial fracture" Is this what you mean? Is not clear what initiates the fracture.

Line 59: "minimum principal stress is close to be aligned to the borehole axis"

Line 65: "Injection volumes.." You do not mention here the injection rate (in litres per minutes), which is one of the constraining factors as well of the induced seismicity.

Line 67: "at which the breakdown occurs"

Line 70: "treatments (of importance here) it can be considered as the pressure...open. ISIP is thus interpreted as..."

Pg. 3

Line 71: "intended for HF or HS, can generate acoustic emissions and microseismic events" (see my comment n. 6)

Line 75: "regardless of scale" What scale? Time- space-scale?

Line 76: "monitoring has been routinely used.." Check you references. They are all in the past. You cannot use the present tense.

Line 78: you may include: Caffagni et al. 2016
Line 78: "At the other extreme of scale". What scale?

Line 84: "indicate changes in the local stress condition"

Line 87: "controlled by.." Here you may develop on the pumped fluid, the pressure gradient (see Dahm et al. 2010, JGR)

Line 94: "direction sigma 3 (Haering…,) particularly for HF operation (Rutledge…; Zoback et al. 2012 SPE)"

Line 100: "Deichmann et al. 2014; Eaton and Caffagni, 2015, First Break)

Pg. 4

Line 107: " Detailed moment tensor….have shown that most of the induced…with relatively a few..."

Line 111: "is very inefficient in radiating" No. Energy from acoustic emissions is radiated efficiently but with a classical monitoring systems of geophones it is not detectable. Please correct.

Line 113: "Thus,...do not necessarily...themselves, yet they contribute to illuminating the overall plane of fracture growth.."

Line 117: "there are a few..between meter-scale….and the ambient stress.."

Line 121: "though it would be desirable"

Line 135: "water injection" You mean the "injection history"? If so, please revise

Line 135: "Then, we detail our anisotropic...localization method.. The obtained results are then compared to the overcoring stress field observations" Is this what you mean?

**Experiment context and study site**

Pg. 5

Line 164: "Since in-situ stress is a relevant factor driving..." you cannot say that is the "major force", also the fluid-injection effects, e.g., pore pressure diffusion and propagation are important

Line 176: remove "that serves"

Line 178: "yields estimate of the full 3D stress"

Pg. 6

Line 184: "The goal is to characterize the local stress condition" see my comment n. 7

Line 187-188: "It was intended...Pahl et al.." the meaning is clear. Rephrase with better English

Line 196: "HTPF". Never mentioned, please spell it

Micro-seismic monitoring

Line 209: "experiments" Include at least one reference

Pg. 7

Line 215: "accelerometers" remove the Italic format

Line 231: "Signals were digitized with a 32-channel..." Is this what you mean?

Line 237: What is the reason of this "dead-time"? Please provide a few explanation

Pg. 8

Line 256: "this spans only 7 m" What you had in mind is the "array's aperture"? If so, please revise

Line 262: "transverse isotropy" Is this correct? Or it is instead "transverse anisotropy"? You mentioned before the elastic anisotropy approximation of Thompson..

Line 282: "1000 time" Is this a peculiar number? Why not stopping the repetition at 100. What would happen? Please provide a few explanation. Also, what about the computational time? This would be important to reproduce the results.

Line 283: "principal component analysis" You can also include the accepted acronym "PCA".

Pg. 9

Also in this section, it might help the reader to see an image of your procedure, a visualization of your located clusters. You can decide to include an additional figure here.
Also a number of questions raises up, such as are there events co-located? Did you identify repeated slips? You can provide a few explanation based on your results

**Results**

Line 319: "due to the lower noise-levels in the borehole" Have you really checked in the traces if what you argued is true? If so, what are the noise levels or the signal-to-noise-ratio?

Pg. 10

I would move Line 322-331 at the beginning of the Results section page 9. First you describe your HF treatment parameter, then you move to the induced seismicity description

Line 329: "not reached during the break-down cycle". Why? Do you have an explanations? That is interesting

You comment about the injection volume, but I think you need to comment as well more on Figure 2, which brings very important information, that are not described in the manuscript.
I would add:
"Induced events occur mainly nucleating in time in correspondence of the peak in both injection pressure and rate." First question here might be: Why we do not observe events later? Also, is there a preference between pressure and volume in inducing seismicity? It looks not..
"In the HF3 case, instead, events look generating also later with respect to the previously observed vertical propagation. This could be a manifestation of fluid-propagation effects and/or fracturing".

It might help also to plot the injection rate with the seismicity rate, to see what are the minimum levels of the injection to trigger seismicity. Perhaps the results might be combined with Figure 3 or to replace Figure 3.

Line 337 "(see Doescth...)"

Line 342: "the impact of considering anisotropy produces variations in the spatially.."

Pg. 11

Figure 5 is neat. Why fractures grow in one direction? Is there a lateral stress gradient factor (see Caffagni et al. 2016, GJI)? Could you plot the direction of sigma H max, to see if there are mis-alignments?

Line 355-358: An explanation of this tendency could be the "Kaiser effect", as observed for instance in the Cooper Basin, (see Baisch et al. 2009, BSSA). I would include this part with one sentence.

Line 379: attach "wave" and "forms" (waveforms)

Pg. 12

Line 402: remove the second "and"

Line 406: "but also accounting for"

In eq. 2. Ai is in the Fourier domain (frequency) or in time domain? It is not clear..

Line 412-414: Please, clarify this sentence..

Pg. 13

Line 435: replace "accord" with "agreement"

Line 436: replace "Figure 2g-i" (?)  with "Figure 10 c"

Line 438: "even with Mra < -3.5 could be located". Ok, but what is your uncertainty in that range?

Pg. 14

Line 454: "the same source mechanism"

Line 457: "it was not...DC mechanism" Where do you shown this behavior?

Line 465: "of volumetric expansion, likely due to the fluid-injection.."

**Discussion**

Line 480: "The direction of propagation of HF3 was different from the other..it propagated downward. HF3.. HF3 also...differently in the instantaneous shut..(ISIP), which decreased with..."

"with cycle to stabilize...Figure 12" What do you really want to say? Is this significant or a minor factor which can be removed?

Line 485: "deepest measurements (HF1 at 18 m) to…, (HF3 at 8 m)."

Pg. 15

Line 490: Please spell "OPTV", never mentioned earlier.

Line 495: "moved 0.3 m downhole, fluid could be injected in a way that fractures were expected to re-open"

Line 499: "they may have worked...hydrofracture test, since the injected fluid was able to penetrate…
Here I actually would add: "Seismicity starts propagating from the packer but not for the HF3 case"

Line 501: "consistent with the evidence that.." Please also check. It is "decameters" or "meters"? In the mentioned figures, it looks like it is in meters..

Line 502: "The low recovery rate of HF3..either by assuming that the packer acts as a sealer of the created fracture after releasing… , or that fluid flows to the far field..."

Line 510: "ranked 5/5 and 4/5"..? Please provide a short explanation. Not all of your readers knows the overcoring technique in details as you do.

Line 516: "and the fault planes of the HF induced seismicity"

Line 523: remove 0 in "090"

Pg. 16

Line 531: "We have shown that micro-seismic monitoring..has provided essential..to obtain a final stress tensor estimate."

Line 536: "may be due to fractures initiated.."

Line 547: "After the initiation, the fracture gradually re-orients itself to become..to the direction preferred of the principal stress."

Here you may develop arguments including the pressure gradient. See my comment n. 8

Line 556: remove "Once"

Line 558: occurred. This reorientation was not...seismic clouds, and it would seem..."
It would be interesting to know why this did not happen..

Pg. 17

Line 561. Did you compute the reduction of sigma n on the foliation plane due to the injection? I expect this to be very low..

Line 570: "We expect focal mechanisms to be in agreement with the stress field orientation… Hence the variability of the mechanisms, which we observe must be due to.."

Line 577: "associated to fracture propagation. In our case, the observation of DC.. exclude (1)"
No. I do not agree. Please see my comment n. 10.

Pg. 18

**Conclusion**

At the beginning, please, you should insert a sentence that recall the experiment (shortly), or a bit of context. "An experiment at the GTS has been conducted...with the purpose of…"

Line 590: "system to study the..at spatial scale from decimeters to meters. The workflow which we have implemented with standard seismological tools, such as..joint location by station corrections... For other seismological...their uncertainties (e.g., Kwiatek…). In the present case, micro-seismic...proved to be crucial to combine interpretation in the results of the stress..."

Line 599: "intervals. Such patterns have an EW strike and dip…"

Line 601: "deviated significantly from the normal to the seismic.."

Line 603: "discrepancy" Among what? Please clarify the two terms of the discrepancy

Line 611: "It is possible that stress..and pressure leak-off effects..influence.. Our observations..surveys conducted in moderately anisotropic rock. A combination of...is essential to obtain a reliable interpretation of the link between stress field and small scale HF growth."

In this way you use words from the abstract and you close the loop.

Pg. 19

**Appendix**

Eq. 2 is meant to be a sum of the ray path contributions in all the layers that you have considered or not?

Line 627: revise "inverse"

Line 633: A verb is missing in the sentence.. Please revise

Line 640: remove the second "becomes"

Pg. 21

Eq. 13. You mention cos -1. Did you mean the $\arccos(x)$ or the $\sec(x) = 1/\cos(x)$? Please specify to avoid confusion

Pg. 23

References

ASTM (2008) is missing!

Pg. 24

Evans et al. It is 2005a or 2005? Please check in the text and revise

Pg. 25

Hollinger et al. You have written in the text "Hollinger". Please, revise

Jeffrey, 2000. Not clear what is it.. a book or a paper?

Manthei et al. 2003. This reference is missing!

Pg. 26

Martinez-Garzon. Please write correctly this surname in the text and reference

Pine and Batchelor. There is written 2003 but also 1984.. Please revise accordingly

Pg. 27

Rutledge et al. 2004. There is another Rutledge and Phillips, 2004 in the text. Please, revise

Van As and Jeffrey (2000). There is another Van As (2000). Is the same? Please, revise

Warpinski et al. The two dates in the references do not match the date in the text. Please, revise

Thomsen and/or Thomson reference is missing! What date then? 1986 or 1989? Please, revise

Figures

Figure 3: Is the "Injected volume" a cumulative injected volume? If so, it is better to revise the horizontal label

Figure 4: Caption "c) Difference...models. It is shown the station.."

Figure 11: Caption: "agrees with one of the focal planes.."

Figure 13: Caption: "Comparison between the foliation plane, fractures….with the seismicity cloud directions..."

---

## Author Comment (AC3) · 21 Nov 2017

**Response to comments by Enrico Caffagni**

**Major concerns**

1. The Abstract is quite long in comparison to standard EGU Discussions abstracts.

*We shortened the abstract by removing the information on fracture initiation at the borehole wall that was probably driven by strength anisotropy, which is not a major overall scope of the paper.*

2. The Discussion can be shortened, avoiding repetitions of concepts.

*We have already shortened the discussion somewhat in response to reviewer 2. However, we feel that there are many aspects of fracture growth that is noteworthy and that we would like to discuss. Thus, we also added a short paragraph discussing the unidirectional fracture growth (referring to Dahm et al., 2010) as suggested by the reviewer further below.*

3. Some references are missing or written in an incorrect way.

*We corrected this.*

4. Table 2 is present, but never mentioned in the text.

*We added reference to Table 2*

5. Figure 11 appears with overlapped labels.

*Corrected.*

6. I feel that a bit of clarification is needed when the authors described the "Microseismic monitoring". I think it is more a question of terminology. The classical induced microseismic events range in frequency between 80 to 300-400 Hz. No more. Sure that the space-time scale is different. Yet you measure events at magnitude -2.5 for instance at KHz. Should we define these as "acoustic emissions"? You actually mention, page 6 line 208-209, "similar to those commonly used in laboratory acoustic emission". For coherency, you should continue to call such events as acoustic emissions, throughout the paper.

*We are aware that typically high frequency seismic events are referred to as acoustic emissions. However, the distinction to seismic events is not clear and different frequency thresholds are given, above which events are supposed to be called acoustic emission. In fact, also events above ~20 Hz are audible and might be referred to as acoustic emissions. In our view, microseismic or just seismic is the more general term describing simply elastic waves. We thus prefer this term over acoustic emission.*

7. I am not sure that your stress characterization study is in the 'far-field' domain, page 6 line 184, at least if this is only due to the "several tens of meter away from any fault". What you are doing, I believe, instead is a characterization of the "local" stress field. Which is more useful.

*We agree that the term 'far-field' is misleading in this context. We changed the term to indicate that it is an estimate of the unperturbed (or less perturbed) stress field some distance away from the fault.*

8. It would benefit your paper to include, at a qualitative way, one or two sentences on the topic: what drives the initiation of an hydrofracture, and "when" or "why" it decides to stop? I suggest to refer to Dahm et al. (JGR, 2010), who discuss the effect of the pore pressure gradient, or pressure gradient, which in your experiment might be also responsible of fracture plane deviation or the trend in asymmetrical growth, page 16 line 544 better. Your pressure gradient might drive the fracture, but

when this starts decreasing, well, your fracture "feels" the local stress field then re-orients itself naturally. Another plausible explanation of fracture plane rotation is a high resilient tectonic stress, see Cooke et al. (2016, TLE). (it is just an attempt of clarification; there are better interpretation; work on that)

*The work by Dahm et al., (2010) is indeed a possible explanation for our asymmetric fracture growth (although possibly not a sufficient explanation). In case of the work by Cooke et al., (2016), we are not convinced that it is transferable to our case.*

*We added a section discussing the asymmetric fracture growth observed based on Dahm et al., (2016)*

9. I honestly do not see a strong connection between polarization through stacking and focal mechanisms. In Figure 11 in a few beach-balls a few events appear in different color (black and white) yet in the same section of the ball. This seems to be contradictory. Actually, by itself, the stacking operations is for enhancing the signal arrival, not for polarization; unless one projects first the traces into already obtained polarization vectors then perform the stacking. By such methods one can locate microseismic events (see Caffagni et al. 2016, GJI).

*We agree that the waveform stacking to retrieve better polarization analysis is problematic and have removed this from the text and the corresponding figure (also in response to reviewer 2).*

Beside that, the lack in the majority of strike-slip events or a combination of different source mechanisms seems to be a kind of constant in induced seismicity. You can see and refer for instance to Baisch et al. (2015, BSSA; Figure 12).

*Indeed, other authors have also observed focal mechanisms that deviate from the prevailing stress field, although in most of these cases (Baisch et al., 2015; Deichmann et al., 2014) the majority of the focal mechanisms does agree with the stress field.*

*We added a sentence mentioning these observations by other authors.*

10. The authors need to be careful when they declare "the observation of DC components..exclude 1" page 17 line 577-578. No, I do not agree. A tensile source mechanism has a DC component as well. In addition Figure 8 reproduces what to me is a tensile event, Cluster 2 (max P wave is bigger than S wave). If I had to sort hundreds of events automatically and visually, I would classify that event as a tensile one.

*We disagree on this comment, pure tensile sources do not have a DC component, otherwise they are not pure tensile sources as they require a shear component. Also, a P-wave that is stronger than the S-wave is not necessarily indicative of a tensile event: depending on where you are in the radiation pattern, the P-wave may be much stronger than the S-wave for a pure DC event (i.e. at 45° from the focal planes). In addition, the sensitivity to S- and P-wave in dependence of sensor orientation is not known. Generally, simple visual assessment of S to P-wave ratios may be misleading to distinguish DC from tensile sources; for this full moment tensor analysis is recommended, which is currently not possible with our data.*

11. The authors mention in the Appendix, page 20 line 652, "a solid angle is also referred to as the takeoff-angle". The classical take-off-angle is not defined as a solid angle, see Stein and Wysession page 222.

*We changed the wording to say that the solid angle is a function – or is representative - of the so-called take-off angle.*

12. More than else, since you as ETH group are currently leading the laboratory experimental research of hydraulic fracturing in Europe, it would benefit to develop on the topic: What's for? Switzerland

might be soon venue of massive usage of geothermal exploitation. What can we learn at macro-scale from such experiments at micro-scale? Can just we simply "upscale" our results in cases of real large-scale stimulations? Perhaps the answer is yes. Injection values in pressure are much higher though and perhaps the effect of small-scale rock anisotropy might vanishes in comparison to big deformations due to pore-elastic effects or fluid diffusion or fracturing.

The paper you want to publish is called "EGU Discussion". It would be appropriate if you could "unbalance" yourself and make some qualitative or even quantitative declarations on future perspectives from such mini-scale hydro-fracturing experiments.

*As the reviewer has already indicated that the discussion is rather long, we prefer to refrain from expanding it towards possible implications for experiment several scales larger than our hydrofractures. Although we agree that various observations may have implications for the large reservoir scale, we prefer to discuss our observations at their scale and leave it to the reader to transport insights to the larger-scale.*

In the following you will find my minor concerns. Please, do not reply in your response file to all of them. But make sure to read them and revise. I have also added additional important concepts missing.

**Minor concerns**

**Abstract**

Line 15: "transverse" to what? To the radial? Usually P-waves propagates in the radial direction

*'Transverse isotropic' is a standard term in anisotropic elasticity theory. Transverse possible related to the symmetry axis.*

Line 20: " from the overcoring stress. An anisotropic elastic model.."

*We changed it to '...overcoring stress tests, provided an anisotropic …'. By adding a comma it becomes clear the second part after the comma belongs to the first part of the sentence.*

Line 21 "sigma1 is significantly.." to Line 24 "the north". Is not clear and I would simply remove it, to shorten the abstract

*The stress field characteristics described here are key outcomes of the paper. We prefer to mention it in the abstract. We shortened the abstract by removing aforementioned information (Comment 1 above)*

**Introduction**

Line 37: "Hydraulic fracturing or hydrofracking (HF)"

*We prefer to avoid the more colloquial term 'hydrofracking'.*

Line 37: Rephrase "Hydraulic fracturing induces artificial fracture networks in a rock mass by highpressure fluid-injection. It has become an essential technique…, for instance to enhance the permeability…, and increase… . HF should not be confused with hydroshearing (HS). HS is a method of rock.. that uses fluid-injection….promoting shear failure, attendant dilation of pre-existing fractures, and fault slippage…"

*We partially adapted the changes in the manuscript.*

Line 45: "criticality of the discontinuity sets" What do you mean? It is not clear

*We added 'proximity to failure' as explanation.*

Line 47: "HS has been often exploited in… . HF small volume has been also utilized in stress.." I would replace the "deep geothermal projects" with the EGS, the "Enhanced Geothermal systems"

*We use the acronyms as suggested and use enhanced geothermal systems.*

Line 52: "etc; see Zang and...2010"

*Changed.*

Line 52: "To better constrain the stress field...and sections where no pre-existing fractures are present"

*We prefer to keep our formulation for brevity.*

Line 55: "water is injected at a constant rate until the ..down, initiating a fracture at the borehole wall...sub parallel to the principal…., and significant deviations are not expected due to the tensile strength anisotropy.." Is this what you mean with "no complications from tensile..." If not, please provide explanations.

*Changed as suggested.*

Line 57: ", then high-pressure fluid injected will tend to initiate an axial fracture" Is this what you mean? Is not clear what initiates the fracture.

*Changed as suggested.*

Line 59: "minimum principal stress is close to be aligned to the borehole axis"

*Our formulation is correct.*

Line 65: "Injection volumes.." You do not mention here the injection rate (in litres per minutes), which is one of the constraining factors as well of the induced seismicity.

*We here refer to the size of the hydro-fracture, which is constraint be volume and not injection rate.*

Line 67: "at which the breakdown occurs"

*Changed.*

Line 70: "treatments (of importance here) it can be considered as the pressure...open. ISIP is thus interpreted as..."

*Changed partially as suggested.*

Line 71: "intended for HF or HS, can generate acoustic emissions and microseismic events" (see my comment n. 6)

*We prefer not to distinguish between acoustic emissions and microseismicity as the distinction is not clear.*

Line 75: "regardless of scale" What scale? Time- space-scale?

*We refer to the HF scale.*

*We added this in the text.*

Line 76: "monitoring has been routinely used.." Check you references. They are all in the past. You cannot use the present tense.

*Using the past is also not appropriate as microseismicity is still being used routinely.*

Line 78: you may include: Caffagni et al. 2016

*We added the reference.*

Line 78: "At the other extreme of scale". What scale?

*We refer to the HF scale.*

*We added this in the text.*

Line 84: "indicate changes in the local stress condition"

*We prefer to keep our formulation as it is more complete.*

Line 87: "controlled by.." Here you may develop on the pumped fluid, the pressure gradient (see Dahm et al. 2010, JGR)

*Here we only discuss impact of stress field and anisotropy on fracture orientation. However, we later added in the discussion section a discussion asymmetric fracture growth that includes the interesting study by Dahm et al., (2010).*

Line 94: "direction sigma 3 (Haering…,) particularly for HF operation (Rutledge…; Zoback et al. 2012 SPE)"

*Changed as suggested.*

Line 100: "Deichmann et al. 2014; Eaton and Caffagni, 2015, First Break)

*We added the reference as suggested.*

Line 107: " Detailed moment tensor….have shown that most of the induced…with relatively a few..."

*Changed partially.*

Line 111: "is very inefficient in radiating" No. Energy from acoustic emissions is radiated efficiently but with a classical monitoring systems of geophones it is not detectable. Please correct.

*We did not claim that acoustic emissions are inefficient in radiating energy, but tensile fractures. The notion is also supported by the fact that acoustic emissions mostly have double-couple sources, and only rarely are secondary opening components observed.*

Line 113: "Thus,...do not necessarily...themselves, yet they contribute to illuminating the overall plane of fracture growth.."

*We feel our formulation is more precise.*

Line 117: "there are a few..between meter-scale….and the ambient stress.."

*Changed as suggested.*

Line 121: "though it would be desirable"

*Our formulation is correct, too.*

Line 135: "water injection" You mean the "injection history"? If so, please revise

*Corrected.*

Line 135: "Then, we detail our anisotropic...localization method.. The obtained results are then compared to the overcoring stress field observations" Is this what you mean?

*Changed as suggested.*

Line 164: "Since in-situ stress is a relevant factor driving..." you cannot say that is the "major force", also the fluid-injection effects, e.g., pore pressure diffusion and propagation are important

*Changed as suggested.*

Line 176: remove "that serves"

*Changed.*

Line 178: "yields estimate of the full 3D stress"

*Changed.*

Line 184: "The goal is to characterize the local stress condition" see my comment n. 7

*Changed accordingly.*

Line 187-188: "It was intended...Pahl et al.." the meaning is clear. Rephrase with better English

*We changed the sentence to make it clearer.*

Line 196: "HTPF". Never mentioned, please spell it

*Corrected.*

Line 209: "experiments" Include at least one reference

*We added a reference.*

Line 215: "accelerometers" remove the Italic format

*Changed (also for piezoelectric sensors further above).*

Line 231: "Signals were digitized with a 32-channel..." Is this what you mean?
*Changed.*

Line 237: What is the reason of this "dead-time"? Please provide a few explanation

*We explain in brackets that this is related to the system being occupied to store the detected waveform.*

Line 256: "this spans only 7 m" What you had in mind is the "array's aperture"? If so, please revise

*Changed.*

Line 262: "transverse isotropy" Is this correct? Or it is instead "transverse anisotropy"? You mentioned before the elastic anisotropy approximation of Thompson..

*Yes, 'transverse isotropy' is correct and a standard term in literature.*

Line 282: "1000 time" Is this a peculiar number? Why not stopping the repetition at 100. What would happen? Please provide a few explanation. Also, what about the computational time? This would be important to reproduce the results.

*We add a short explanation that 1000 repetitions yields a statistically representative sample.*

Line 283: "principal component analysis" You can also include the accepted acronym "PCA".

*We prefer to spell out for clarity.*

Also in this section, it might help the reader to see an image of your procedure, a visualization of your located clusters. You can decide to include an additional figure here. Also a number of questions raises up, such as are there events co-located? Did you identify repeated slips? You can provide a few explanation based on your results

*We here limit ourselves to the method description, while the results on located clusters and repeated events are covered later.*

Line 319: "due to the lower noise-levels in the borehole" Have you really checked in the traces if what you argued is true? If so, what are the noise levels or the signal-to-noise-ratio?

*We indicate that the noise-level is less than half of the one from the tunnel sensors.*

I would move Line 322-331 at the beginning of the Results section page 9. First you describe your HF treatment parameter, then you move to the induced seismicity description

*We prefer to keep it as it is, as moving this section would compromise the text structure. We first describe general event statistics (number of detected/located events) and then describe when they occurred (breakdown cycles, reopening cycles or after shut-in).*

Line 329: "not reached during the break-down cycle". Why? Do you have an explanations? That is Interesting

*Unfortunately, we do not have an explanation for it, and we do not want to expand the discussion towards this topic. We prefer to simply report this observation for future research to address the it.*

You comment about the injection volume, but I think you need to comment as well more on Figure 2, which brings very important information, that are not described in the manuscript.
I would add:
"Induced events occur mainly nucleating in time in correspondence of the peak in both injection pressure and rate." First question here might be: Why we do not observe events later? Also, is there a preference between pressure and volume in inducing seismicity? It looks not.

*We do not observe that seismicity only occurs when pressure or rate are at their peaks. In fact the largest seismicity rate occurs somewhat after the peak pressures during breakdown or reopening.*

*However, we agree that it is interesting to state that seismicity rate somewhat depend on injection rate but not on pressure.*

*We added a statement describing these observations in the manuscript.*

"In the HF3 case, instead, events look generating also later with respect to the previously observed vertical propagation. This could be a manifestation of fluid-propagation effects and/or fracturing".
It might help also to plot the injection rate with the seismicity rate, to see what are the minimum levels of the injection to trigger seismicity. Perhaps the results might be combined with Figure 3 or to replace Figure 3.

*We do not follow the observation regarding HF3. Post-shut-in seismicity also occurs in the cases of HF1 and HF2. Clearly, seismicity rates and propagation reflects fluid-propagation and fracturing. We feel it is not necessary to additionally comment on this in the text.*

*Also as mentioned above, injection pressure and rate-dependence of seismicity can be observed from Figure 2 and does not require an additional figure.*

Line 337 "(see Doescth...)"

*We added the word 'see'.*

Line 342: "the impact of considering anisotropy produces variations in the spatially.."

*We believe our formulation is more precise.*

Figure 5 is neat. Why fractures grow in one direction? Is there a lateral stress gradient factor (see Caffagni et al. 2016, GJI)? Could you plot the direction of sigma H max, to see if there are misalignments?

*Currently, we do not have an explanation for the downward or upward propagation. The comparison with the stress field estimates from overcoring is done later in Section 5.2 and Figure 13. Note that it does not make sense to discuss the stress field in terms of sigmaHmax, as the stress tensor is rotated with respect to ground surface or the horizontal.*

Line 355-358: An explanation of this tendency could be the "Kaiser effect", as observed for instance in the Cooper Basin, (see Baisch et al. 2009, BSSA). I would include this part with one sentence.

*We added a sentence on this.*

Line 379: attach "wave" and "forms" (waveforms)

*Corrected.*

Line 402: remove the second "and"

*Corrected.*

Line 406: "but also accounting for"

*Our formulation is correct.*

In eq. 2. Ai is in the Fourier domain (frequency) or in time domain? It is not clear..

*We added that it is in time-domain.*

Line 412-414: Please, clarify this sentence.

*We understand that the reviewer has read the originally submitted version of the manuscript. The updated manuscript addressing comments from Reviewer 2, already addressed this sentence, too.*

Line 435: replace "accord" with "agreement"

*Changed.*

Line 436: replace "Figure 2g-i" (?) with "Figure 10 c"

*We are actually referring to Figure 2g-i in this sentence and not to Figure 10c.*

Line 438: "even with Mra < -3.5 could be located". Ok, but what is your uncertainty in that range?

*We indicate in the text that these magnitudes were located with an error better than 2 m.*

Line 454: "the same source mechanism"

*This part was shortened in response to reviewer 2.*

Line 457: "it was not...DC mechanism" Where do you shown this behavior?

*This part was shortened in response to reviewer 2.*

Line 465: "of volumetric expansion, likely due to the fluid-injection.."

*This part was shortened in response to reviewer 2.*

Line 480: "The direction of propagation of HF3 was different from the other..it propagated downward. HF3.. HF3 also...differently in the instantaneous shut..(ISIP), which decreased with..." "with cycle to stabilize...Figure 12" What do you really want to say? Is this significant or a minor factor which can be removed?

*We prefer to keep our formulation. We here only state our observations. We prefer to not comment on their significance although we believe that they are significant as they are consistent throughout all cycles.*

Line 485: "deepest measurements (HF1 at 18 m) to…, (HF3 at 8 m)."

*Changed as suggested.*

Line 490: Please spell "OPTV", never mentioned earlier.

*Changed as suggested.*

Line 495: "moved 0.3 m downhole, fluid could be injected in a way that fractures were expected to reopen"

*Partially changes as suggested*

Line 499: "they may have worked...hydrofracture test, since the injected fluid was able to penetrate… Here I actually would add: "Seismicity starts propagating from the packer but not for the HF3 case"

*We prefer to keep our formulation, which we believe is clearer.*

Line 501: "consistent with the evidence that.." Please also check. It is "decameters" or "meters"? In the mentioned figures, it looks like it is in meters..

*In fact, it is decimetres. We changed this in the text.*

Line 502: "The low recovery rate of HF3..either by assuming that the packer acts as a sealer of the created fracture after releasing… , or that fluid flows to the far field..."

*We prefer to keep our formulation.*

Line 510: "ranked 5/5 and 4/5"..? Please provide a short explanation. Not all of your readers knows the overcoring technique in details as you do.

*We added a short explanation.*

Line 516: "and the fault planes of the HF induced seismicity"

*'Fault' is not an appropriate term for fracture at a scale of meters. We prefer to keep our formulation.*

Line 523: remove 0 in "090"

*Changed.*

Line 531: "We have shown that micro-seismic monitoring..has provided essential..to obtain a final stress tensor estimate."

*Changed as suggested.*

Line 536: "may be due to fractures initiated."

*Changed.*

Line 547: "After the initiation, the fracture gradually re-orients itself to become..to the direction preferred of the principal stress."

*We keep our formulation as it is more precise.*

Here you may develop arguments including the pressure gradient. See my comment n. 8

*We added a short discussion on asymmetric fracture growth and pressure/stress gradients.*

Line 556: remove "Once"

*Changed.*

Line 558: occurred. This reorientation was not...seismic clouds, and it would seem..."
It would be interesting to know why this did not happen..

*We keep our formulation.*

Line 561. Did you compute the reduction of sigma n on the foliation plane due to the injection? I expect this to be very low..

***The goal of this calculation is to compare $\sigma_n$ on the foliation plane to $\sigma_3$. If the injection pressure is used to compute an effective stress, both quantities would reduce by the same amount. Thus, we do not see how this changes the discussion.***

Line 570: "We expect focal mechanisms to be in agreement with the stress field orientation… Hence the variability of the mechanisms, which we observe must be due to.."

***The section has changed in response to comments by reviewer 2.***

Line 577: "associated to fracture propagation. In our case, the observation of DC.. exclude (1)"
No. I do not agree. Please see my comment n. 10.

***Double-couple events – even if not pure and imposed by a tensile component – do require shear motion along the source plane. Such a mechanism is not in agreement with pure tensile fracture opening.***

***We reworded this sentence to explicitly state that pure tensile fracturing can be excluded for the double-couple events (but not necessarily in general).***

Conclusion
At the beginning, please, you should insert a sentence that recall the experiment (shortly), or a bit of context. "An experiment at the GTS has been conducted...with the purpose of…"

***We added such an introductory sentence.***

Line 590: "system to study the..at spatial scale from decimeters to meters. The workflow which we have implemented with standard seismological tools, such as..joint location by station corrections... For other seismological...their uncertainties (e.g., Kwiatek…). In the present case, micro-seismic...proved to be crucial to combine interpretation in the results of the stress..."

***We partially changed the formulation for better clarity.***

Line 599: "intervals. Such patterns have an EW strike and dip…"

***We keep our formulation.***

Line 601: "deviated significantly from the normal to the seismic.."

***We keep our formulation.***

Line 603: "discrepancy" Among what? Please clarify the two terms of the discrepancy

***It is the discrepancy explained in the previous sentence. We reworded the previous sentence to make it clearer.***

Line 611: "It is possible that stress..and pressure leak-off effects..influence.. Our observations..surveys conducted in moderately anisotropic rock. A combination of...is essential to obtain a reliable interpretation of the link between stress field and small scale HF growth."

In this way you use words from the abstract and you close the loop.

***We prefer to use an alternative formulation for the conclusion.***

**Appendix**

Eq. 2 is meant to be a sum of the ray path contributions in all the layers that you have considered or not?

*We state that it is the entire ray path length.*

Line 627: revise "inverse"

*Corrected.*

Line 633: A verb is missing in the sentence.. Please revise

*Corrected.*

Line 640: remove the second "becomes"

*Corrected.*

Eq. 13. You mention cos -1. Did you mean the arccos(x) or the sec(x) =1/cos(x)? Please specify to avoid confusion

*We meant arccos and corrected it.*

References

ASTM (2008) is missing!

*We could not find the reference ASTM (2008) so we changed the citation to Zang and Stephansson, (2010)*

Evans et al. It is 2005a or 2005? Please check in the text and revise

*We removed the a.*

Hollinger et al. You have written in the text "Hollinger". Please, revise

*It should be Holliger. We changed it in the text.*

Jeffrey, 2000. Not clear what is it.. a book or a paper?

*It is actually a patent. We reference it properly.*

Manthei et al. 2003. This reference is missing!

*No, it is actually there.*

Martinez-Garzon. Please write correctly this surname in the text and reference

*We corrected it in the text.*

Pine and Batchelor. There is written 2003 but also 1984.. Please revise accordingly

*It should be 1984. Corrected.*

Rutledge et al. 2004. There is another Rutledge and Phillips, 2004 in the text. Please, revise

*Rutledge and Phillips, 2004 was changed into Rutledge et al, 2004 in the text.*

Van As and Jeffrey (2000). There is another Van As (2000). Is the same? Please, revise

*Van As (2000) was changed into Van As and Jeffrey (2000).*

Warpinski et al. The two dates in the references do not match the date in the text. Please, revise

*Corrected.*

Thomsen and/or Thomson reference is missing! What date then? 1986 or 1989? Please, revise

*We added the reference and corrected the year. It is Thomson, 1986.*

Figure 3: Is the "Injected volume" a cumulative injected volume? If so, it is better to revise the horizontal label

*We corrected it in the caption.*

Figure 4: Caption "c) Difference...models. It is shown the station.."

*Changed as suggested.*

Figure 11: Caption: "agrees with one of the focal planes.."

*The figure and caption has changed in response to reviewer 2.*

Figure 13: Caption: "Comparison between the foliation plane, fractures….with the seismicity cloud directions..."

*We keep our formulation.*

---

## Author Comment (AC4) · 21 Nov 2017

[revised manuscript text omitted]

500    ~~cluster that give rise to slight change in radiation direction. Although the polarization pattern is comparable to the patterns from single events, it was not possible to fit a double-couple mechanism. Currently, we cannot explain the discrepancy. Possibly the mechanisms of the events within the cluster vary (as the variable S/P wave amplitude ratios also indicate; Figure 8), and thus waveform stacking does not give meaningful polarizations. Another reason may be that there are components of non-~~

505    It is noteworthy that the normal faulting style contradicts

the stress field observations. As described in Section 5.2, the maximum principal stress $\sigma_1$ and $\sigma_3$ are sub-horizontal and $\sigma_2$ and $\sigma_3$ are very close in magnitude suggesting that a thrust fault to strike-slip mechanism is expected.  Note that in many other induced seismicity studies most focal mechanisms were in agreement with the prevailing stress field, with only few events deviating from it (e.g., Baisch et al., 2015; Deichmann et al., 2014). Possibly, in our case, a component of volumetric expansion or a compensated linear vector dipole (CLVD) mechanism modifies the pure double-couple mechanisms (Vavrycuk, 2011, Martínez-Garzón et al., 2017). Volumetric expansion would be consistent with growth of a tensile fracture driven by fluid injection.

The double-couple mechanisms found here are in agreement with many studies that showed that seismicity associated with HF have double-couple sources (Chitrala et al., 2013; Dahm et al., 1999; Nolan-Hoeksema & Ruff, 2001; Ohtsu, 1991). ~~The events are thought to be induced either by pressure propagating into the small pre-existing fractures adjacent to growing HF (e.g. Rutledge and Phillips, 2004) or by stress changes induced around the propagating HF (as suggested by Nolan-Hoeksema & Ruff, 2001). In the latter case, it is expected that the stress field locally rotates and that deviations from the ambient stress field reflected in the focal mechanisms occur. Future analysis of seismicity of similar experiment performed at the GTS may give more insights into this.~~

[revised manuscript text omitted]
 also moreit permits structures that are misnot optimally-oriented in the stress field canto be reactivated. ButIt is also possible that stress field perturbations arising from the propagating hydrofractures (mechanism 3) may additionally promote criticality along these planes, although this cannot explain normal and thrust faulting events in close proximity. The observationis not resolved by the present observations. In this regard, the assumption of the latter would seem to indicate severe stress field perturbation around the sources may be involved, as in mechanism (3) above. As the principle stress magnitudes are relatively close in our case (possibly < 5 MPa between $\sigma_1$ and $\sigma_3$) such stress perturbation may lead to a local changestress homogeneity within the study volume inherent in the stress regime. Future work is

required to reveal the dominant mechanism leading to observed double-couple seismicity. 
[revised manuscript text omitted]

1155

1160

---

## Author Comment (AC5) · 21 Nov 2017

The comment was uploaded in the form of a supplement:
https://www.solid-earth-discuss.net/se-2017-78/se-2017-78-AC5-supplement.pdf

---

## Author Response (AR2)

**Response to comments by Topical Editor Charlotte Krawczyk**

*Thank you for your positive feedback. We addressed all comments as recommended.*

1) rev1, comment 1: is that, what you answer in the rebuttal letter said somewhere in the text ? I'd prefer to see a short phrase on this issue, but am not sure where to find it.

I assume it is the following comment by reviewer 2 (as reviewer 1 did not have any comments):

The Authors put a lot of efforts in constraining the hypocenters of AE activity. However, I am puzzled why they did not use the S-waves to improve the location quality? From Figure and the paper itself it is clear the S-waves were efficiently recorded and they could help to constrain the locations. S-phases have been applied in previous studies using similar acquisition system with success (JAGUARS project, ASPO FHF experiment, see appropriate papers). Could you comment on that and also inform the Readers on your choices?

*We added the sentence: 'S-wave arrivals were not included in the location, because only few S-wave arrivals could be reliably picked, and the anisotropic S-wave velocity model is not well constrained.' in the methods section (Line 265-266).*

2) rev3, comment 10: the disagreement is fine with me.

*We are glad you support our argument.*

3) rev3, line 329: your rebuttal is not included in the text; please add a statement on this issue, too.

The comment is: "not reached during the break-down cycle". Why? Do you have an explanations? That is Interesting

*We added a statement that we do not understand the reason for such a minimum volume threshold to induce seismicity (Line 345-346).*